# Automated detection and classification of synoptic scale fronts from atmospheric data grids

Stefan Niebler[1], Annette Miltenberger[2], Bertil Schmidt[1], and Peter Spichtinger[2]

[1]Institut für Informatik, Johannes Gutenberg-Universität Mainz, Staudingerweg 7, 55128 Mainz, Germany
[2]Institut für Physik der Atmosphäre, Johannes Gutenberg-Universität Mainz, Becherweg 21, 55128 Mainz, Germany

**Correspondence:** Stefan Niebler (stnieble@uni-mainz.de)

**Abstract.** Automatic determination of fronts from atmospheric data is an important task for weather prediction as well as for research of synoptic scale phenomena. In this paper we introduce a deep neural network to detect and classify fronts from multi-level ERA5 reanalysis data. Model training and prediction is evaluated using two different regions covering Europe and North America with data from two weather services. We apply label deformation within our loss function which removes the need for skeleton operations or other complicated post processing steps as observed in other work, to create the final output. We observe good prediction scores with Critical Success Index higher than $66.9\%$ and a Object Detection Rate of more than $77.3\%$. Frontal climatologies of our network are highly correlated (greater than $77.2\%$) to climatologies created from weather service data. Comparison with a well-established baseline method based on thermodynamic criteria shows a better performance of our network classification. Evaluated cross sections further show that the surface front data of the weather services as well as our networks classification are physical plausible. A study linking fronts to extreme precipitation events is conducted to showcase possible applications of the proposed method. This demonstrates the use of our new method for scientific investigations.

## 1 Introduction

Atmospheric fronts are ubiquitous structural elements of extra-tropical weather. The term *front* refers to a narrow transition region between airmasses of different density and/or temperature (see, e.g., Thomas and Schultz, 2019b). These airmass boundaries play an important role for understanding the dynamics of midlatitude weather, since they are usually related to clouds. Fronts are often associated with significant weather, such as intense precipitation and high gust speeds. Hence, fronts in the sense of separating polar from more subtropical airmasses played a vital part of the communication of weather to the public and the public perception of weather in general, although this aspect may have lost some attention due to the use of colourful apps. Frontal surfaces exist also on smaller scales, e.g. in the context of sea-breeze circulation or local circulation patterns in mountainous regions. Even tropical weather systems might indeed produce similar features of transition regions of different

airmasses, but due to other reasons than in the extra-tropical weather systems. However, the focus here and in much of the lit-
erature is on the larger-scale fronts that can extend over several hundred kilometres and are often associated with extra-tropical
cyclones (Schemm et al., 2018). Quasi-stationary fronts exist that can extend over large distance and do typically not move
strongly over time, e.g. the Mei-Yu front (e.g. Hu et al., 2021). These stationary fronts are as well foci of significant surface
weather. Unfortunately, there is no generally accepted front definition, see, e.g. the discussion in Schemm et al. (2018) and
Thomas and Schultz (2019a). Thus, the detection of fronts often rely on different measures, usually based on physical vari-
ables and including physical hypotheses or theories. Additionally, it is still on debate if a front detection should be guided by
determining surface fronts (as, e.g., on the analysis charts of weather services), or even more on the physical (horizontal and
vertical) structure (see also the summary in Uccellini et al., 1992; Sanders, 1999).

Nevertheless, determining the position and propagation of surface fronts plays an important role for weather forecasting,
and, of course, for research on synoptic scale phenomena. The traditional manual approach to front detection is based on
the expertise of weather analysts at operational meteorological services, along some (even empirical) guidelines. With the
advent of large, gridded reference data-sets, e.g. reanalysis from different weather centres, as e.g. ECMWF or NCEP, in the
second half of the past century the drive for objective means to detect fronts automatically set in (see, e.g., Hewson and
Titley, 2010). Currently used methods are typically relying on detecting strong gradients in either temperature and humidity
fields (e.g., by using equivalent potential temperature or wet-bulb temperature) or in wind fields (Schemm et al., 2015). The
former methodology goes back to the work by Renard and Clarke (1965) and is represented by Hewson (1998), who suggested
an automatic method to detect fronts in fairly coarse data sets based on the so-called "thermal front parameters", derived
from thermodynamic variables. In these and subsequent studies this is often related to the second spatial derivative of the
temperature, and one or more "masking parameters", i.e. thresholds of thermal gradients along the front or in adjacent regions.
This or conceptually similar methods have been used in numerous studies to determine the global or regional climatological
distribution of fronts (e.g. Berry et al., 2011; Jenkner et al., 2010).

For the investigation of fronts on the southern hemisphere Simmonds et al. (2012) suggested an alternative approach that
investigates the Eulerian time rate of change of wind direction and speed in the lower troposphere at a given location. A
comparison of the two methods to identify fronts on a global climatological scale by Schemm et al. (2015) revealed some
agreement between the fronts detected, but also regional difference and systematic biases in the detection of certain front
types by both algorithms: For example, the "thermal" method detects more reliably warm fronts than the method based on
lower tropospheric wind speed and direction. In addition, the orientation of detected fronts differs in general between the two
methods. In consequence Schemm et al. (2015) also find differences in the global distribution of fronts and the amplitude of
seasonal variations in front occurrence frequency.

While it is well known that different front detection methods provide different outputs (e.g. Schemm et al., 2015; Hope et al.,
2014), an objective ground-truth is difficult to find. Most studies developing or testing automatic detection schemes rely on
manual analysis as the "gold standard" to test the accuracy and for tuning free parameters in the automatic detection schemes
(e.g., Hewson, 1998; Berry et al., 2011; Bitsa et al., 2019). However, it should be noted that manual analysis is affected to a
large degree by subjective decisions, and hence the focus, interest and expertise, of the person conducting the analysis. Shakina

(2014) reports results from an inter-comparison study of different manual front analysis carried out independently in different divisions of the Russian Meteorological service up until the 1990s. Comparing the different archives agreement on the presence or absence of a front in any one $2.5° \times 2.5°$ box was found in 84.8 % of cases. However, if only the presence of fronts in any one grid box is considered the agreement dropped to 23 % to 30 % depending on the type of front. Shakina (2014) further suggests that disagreement mainly arises from the detection and positioning of secondary or occluded fronts which typically are associated with less marked changes in surface weather. It is likely that the differences between manual analysis by different forecasters in the meantime have not reduced, but they may potentially be reduced by strict guidelines for forecasters on the key decision features for positioning fronts.

Despite a none negligible subjectivity of manual analysis, it still offers many advantages over automatic methods:

1. In contrast to most automatic detection methods many different aspects, including temperature, wind, and humidity fields, surface pressure, but also surface precipitation and wind, are taken into account.

2. Manual analysis does not rely strongly on the choice of (arbitrary) thresholds that are needed in most automatic front detection algorithms.

3. Experience of analysts can be taken into account, especially on regional scales (e.g. with complicated terrain as in the Alps)

In order to address the over-reliance on specific variables some recent studies have suggested methods that combine not only temperature and humidity data but also include information on the wind field (e.g. Ribeiro et al., 2016; Parfitt et al., 2017), or information on Eulerian changes in mean sea-level pressure (e.g. Foss et al., 2017). Nevertheless these extended algorithms that are so far mainly used in regional studies still rely on choosing appropriate thresholds for the magnitude of thermal gradients or changes in the wind direction and speed.

The necessity of manually designing metrics and selecting thresholds for automatic front detection can be at least partly overcome by employing statistical methods and machine learning approaches. The key idea with this approach is that based on manual analysis a complex statistical method retrieves as much consistent information on patterns, important variables, and thresholds as is available in manual analyses and coinciding state of the atmosphere, e.g. from reanalysis data-sets. Previous attempts on using machine learning approaches for front detection are discussed in more detail the following section.

Recently different groups have used Artificial Neural Networks (ANNs) to predict frontal lines from atmospheric data. Biard and Kunkel (2019) used the MERRA-2 data-set to predict and classify fronts over the North American continent. Their network also classifies their predicted fronts using the four types: warm, cold, stationary, and occlusions. They used labels provided by the North American Weather Service.

Lagerquist et al. (2019) used the North American Regional Reanalysis (NARR) data-set (Mesinger et al., 2006), to predict synoptic cold and warm fronts over the North American continent also using the NWS labels. While the network of Biard and Kunkel (2019) creates an output on the input domain, the network of Lagerquist et al. (2019) predicts the probability for a single pixel and needs to be applied to each pixel consecutively. Both methods rely on postprocessing steps like morphological

thinning to create their final representation of frontal data. Additionally, both methods only use a 2D mask for each input variable not making use of multiple pressure or height levels. Matsuoka et al. (2019) used a U-Net architecture (Ronneberger et al., 2015) to predict stationary fronts located near Japan.

In this study we present a new method for automatic front detection based on machine learning, which uses meteorological reanalysis as input data, whereas the method is trained with information on surface fronts as provided by two different weather services. The overall aim of this study is to investigate the degree to which machine learning approaches are able to replicate manual analysis on a case-study and climatological scale and the degree to which the learned features are consistent with meteorological expectations on the physical properties characterising a frontal surface. Our provided network also uses the U-

Net approach to predict and classify all four types of fronts, without the need of morphological post processing. Additionally we evaluate our approach similar to Lagerquist et al. (2019) using an object based evaluation method. Unlike the previous methods we incorporate data from **two** different weather services, the North American Weather Service (NWS) and the German Weather Service (Deutscher Wetterdienst, DWD) and also evaluate on both regions. We additionally compare our predicted fronts against the method developed by Schemm et al. (2015), using thermal front parameters (TFP), as baseline. As input data

for the method, we use the ERA5 reanalysis data (Hersbach et al., 2020) from the European Centre for Medium-Range Weather Forecasts (ECMWF) at a $0.25°$ grid at multiple pressure levels for each variable. This data-set exhibits a higher resolution than the NARR ($32\,\mathrm{km}$ grid) and used MERRA-2 data-set by Biard and Kunkel (2019) ($1°$ grid). Additionally, we used multiple pressure levels to refine our results.

     Although we are aware of the conceptual differences between determining surface fronts and the complex 3D structure

of fronts, we use the surface maps as a ground truth, i.e. as a proxy for the complex structures fronts. However, in the later evaluation it turns out that the detected surface fronts represents the physical properties in a meaningful way.

     In Section 2 we will describe our used network architecture, data and evaluation methods, respectively. In Section 3 we explain our evaluation methods and display our evaluation results on the training and test data set as well as applications in terms of physical properties of fronts and related extreme precipitation events. These results are also discussed. We close the

study with a summary of the study and a short outlook for future improvements as well as further applications of the new method for scientific purposes.

## 2   Materials and Methods

For each spatial grid point our proposed algorithm predicts a probability distribution, describing how likely it is that the point belongs to each of our possible five classes: warm, cold, occlusion, stationary, or background. Our method predicts that

estimate from a 4-dimensional input consisting of multiple channels located on a 3-dimensional multilevel geospatial grid, which was flattened to a 3-dimensional input by combining the atmospheric channel and level dimension. For this task we use a convolutional neural network (CNN) architecture to automatically learn atmospheric features that correspond to the existence of a weather front at spatial grid points. We use a supervised learning approach, in which we provide ground truth data of frontal data sampled from two different weather services (surface fronts). We adjust hidden parameters of the CNN in

order to optimize a loss function measuring the quality of our weather front prediction. CNN architecture and training will be explained in further detail in this section. Our network was implemented, trained, and tested using Pytorch 1.6 (Paszke et al., 2019). Parallel Multi-GPU training was implemented using Pytorch's DistributedParallel package. The provided code was run using Python 3.8.2 and is freely available (see below).

## 2.1 Data

We will briefly describe which channels and gridpoints were used as training input from the ERA5 reanalysis data (Hersbach et al., 2020). Furthermore, we will describe the format of the corresponding label data of fronts obtained from NWS and DWD; in the case of the DWD label data, we additionally describe the generation process of the DWD data

### 2.1.1 ERA5 Reanalysis Data

Our model input consists of a multichannel multilevel spatial grid provided by ECMWFs ERA5 reanalysis data-set. Each
channel denotes a different atmospheric variable, while levels consist of a subset taken from the $L137$ vertical level definition (ECMWF, 2021). Data is represented on a spatial grid with a grid-spacing of $0.25°$ in both latitudinal and longitudinal direction. Since we do not expect to obtain relevant information from high altitude level data, we decided to restrict ourselves to every fourth level within the inclusive interval $[105, 137]$, representing 9 pressure levels between surface pressure and about $700\,\mathrm{hPa}$. This range contains both the ground level information as well as the $850\,\mathrm{hPa}$ pressure level information, both of which are
commonly used to detect fronts. Pressure levels are defined as parameters of an affine transformation of the surface level pressure, which is why we manually added the surface pressure field to the data using the merge operation of the Climate Data Operators (CDO) (Schulzweida, 2019). This allows us to calculate the pressure at each gridpoint and level. We further only use 5 ERA5 multilevel variables as input for our network: temperature ($T$), specific humidity ($q$), zonal wind velocity ($u$, East-West), meridional wind velocity ($v$, North-South), and vertical velocity ($w$), respectively. In addition the surface pressure
(sp) and longitudinal distance per pixel in $\mathrm{km}$ relative to $27.772\,\mathrm{km}$ (kmPerLon) are considered. The distance between two pixel at a certain degree latitude is derived by assuming a spherical shape of the globe and is only used as a single level variable. Surface pressure on the other hand is used to estimate the pressure at each model level using the corresponding level parameter to create another multilevel network input. All resulting data is normalized with respect to a global mean and variance sampled from data of the year 2016. The resulting mean and variance values are listed in Table 1.
While ERA5 reanalysis data is available for the whole globe the available ground truth labels only reside within the analysis region of their corresponding weather services. We therefore cannot use ERA5 data outside these regions. For this reason we decided to restrict our usage of ERA5 data to rectangular subgrids, each of which being completely within the analysis region of its respective weather service.

The extent of these regions is described in Tab. 2 as DWD$_{\text{input}}$ and NWS$_{\text{input}}$. Pixel at the border of our input may lose
critical information to successfully identify a front due to the input crop. As a result detections on the outer $5°$ (20 pixel) of the input domain are not evaluated during training. While the network still outputs these pixel, they do not contain valid detections and should therefore be removed from the evaluation. As a result the effective output region is smaller than the

**Table 1.** Mean and variance of the individual variables used for normalization of input data.

| variable | (unit) | mean | variance (in unit$^2$) |
|:---:|:---:|:---:|:---:|
| T | K | 275.355461 | 320.404803 |
| q | $\mathrm{kg\,kg^{-1}}$ | $5.57926815 \cdot 10^{-3}$ | $2.72627785 \cdot 10^{-5}$ |
| u | $\mathrm{m\,s^{-1}}$ | 1.27024432 | 67.4232481 |
| v | $\mathrm{m\,s^{-1}}$ | 0.10213897 | 43.6244384 |
| w | $\mathrm{Pa\,s^{-1}}$ | $5.87718196 \cdot 10^{-3}$ | $4.77972548 \cdot 10^{-2}$ |
| sp | hPa | 865.211548 | 1494.6063 |
| kmPerLon | km/° | 0.64 | 0.09 |

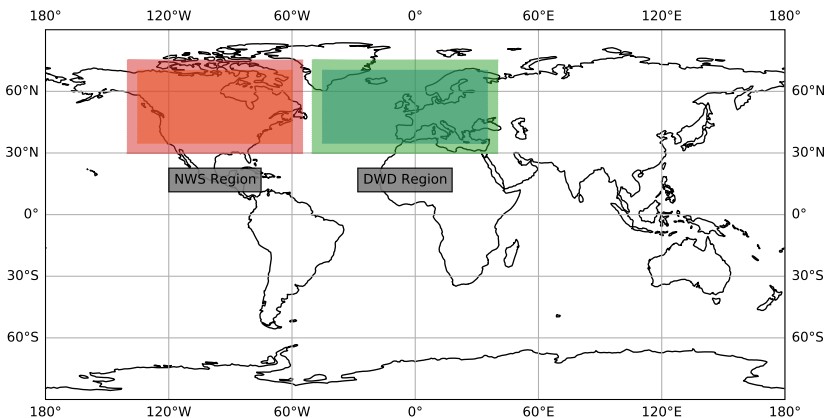

**Figure 1.** Bounding Boxes for the two regions used for training and evaluation against the weather service labels. The brighter area can be used as input, but is not evaluated.

input region, as indicated in tab. 2. This is also shown in Fig. 1 as the difference in shade within each weather service region. Prior to evaluation we create detections for each sample using the global input data. Evaluations against the weather service labels are performed using the corresponding output regions. Comparisons against the baseline method use the same regions restricted to latitudes spanning $[35°, 60°]N$ instead. The evaluation in section 3.2.2 does not rely on the weather service data and is therefore evaluated within $[-60°, 60°]N$ and $[-175°, 175°]E$. The restriction of the longitudes is caused by the smaller output regions, as explained in this section.

### 2.1.2 NWS Front Label Data

For training on the North American continent we use the HiRes Coded-Surface-Bulletins (csb) of the North American National Weather Service (National Weather Service, 2019). This data ranges from 2003 up to 2018 and was previously used by Biard and Kunkel (2019) and Lagerquist et al. (2019). Each front in a csb file consists of an identifier, describing the type of front,

**Table 2.** The used input and output regions for each weather service region during training and the global input region. Levels are only used for network input. The output regions are also used during evaluation against the weather service labels. Every fourth vertical level between levels 105 and 137 is chosen to reduce the amount of input data, also in terms of redundant information.

| Weather Service | Latitudes | Longitudes | Levels | |
|---|---|---|---|---|
| $DWD_{input}$ | $[30°N, 75°N]$ | $[-50°E, 40°E[$ | $[105, 137, 4]$ | |
| $DWD_{output}$ | $[35°N, 70°N]$ | $[-45°E, 35°E[$ | - | |
| $NWS_{input}$ | $[30°N, 75°N]$ | $[-140°E, -55°E[$ | $[105, 137, 4]$ | |
| $NWS_{output}$ | $[35°N, 70°N]$ | $[-135°E, -60°E[$ | - | |
| Global | $]-90°N, 90°N]$ | $[-180°E, 180°E[$ | $[105, 137, 4]$ | |

followed by a series of coordinate pairs on a $0.1°$ grid, defining a polyline of the front. We do not perform any pre-processing on this data. In accordance with our available data we restricted the use of the latter to the years 2012 to 2017 using only
snapshots in a 6-hour interval to keep the amount of data balanced compared to the DWD data during training. The NWS data set contains labels for the following front types: warm, cold, occlusion, and stationary fronts, respectively.

### 2.1.3 DWD Front Label Data

For training over Europe and the Northern Atlantic we use label data extracted from the surface analysis maps of the Deutscher Wetterdienst (DWD) for the years 2015 to 2019. Unlike the Coded-Surface-Bulletins, these maps are not provided as polylines,
but rather as a PNG images of a region containing both the North Atlantic as well as Western Europe (see Fig. 2 (a)). Each of those images has a resolution of $4389 \times 3114$ pixel. To use the labels we extract each individual front, by creating coordinate pairs, which describe the front as a polyline, similar to a csb. Within an image different types of fronts are color coded, which allows us to easily separate them from the background. We remove the symbolic identifiers like half-circles and triangles, indicating the directions of a front, as we do not need this information. Otherwise, these symbols could create false positive
coordinate points in the label data. Our algorithm first filters all fronts of a specific type by filtering all pixel of the corresponding color. In a second step we erase all additional symbols on each line. Subsequently, latitude and longitude coordinate pairs along each line are extracted in order to describe each front in terms of a polyline. In Fig. 2 (b) we show an example of a processed image file, redrawn onto the same projection as the input image. Blue and red lines in both panels correspond to cold and warm fronts respectively, while green lines correspond to occlusions, which are pink in the left panel.
In certain cases our method fails to correctly extract the frontal lines. These cases lead to gaps within a front, wrongly extracted objects or wrongly connected fronts. Gaps originate from two factors. One is that another object is drawn on top of a frontal line, effectively splitting the front into two parts. The other cause of a gap is an aggregation of multiple front-symbols on a short segment. As our method removes sections where a symbol is placed before reconnecting the remaining points, crowded placement of these symbols may make the remaining part of the front too short to be considered relevant and as such will be
omitted. Wrongly extracted objects occur mostly with storms that are depicted in the same color as a warm front. As such

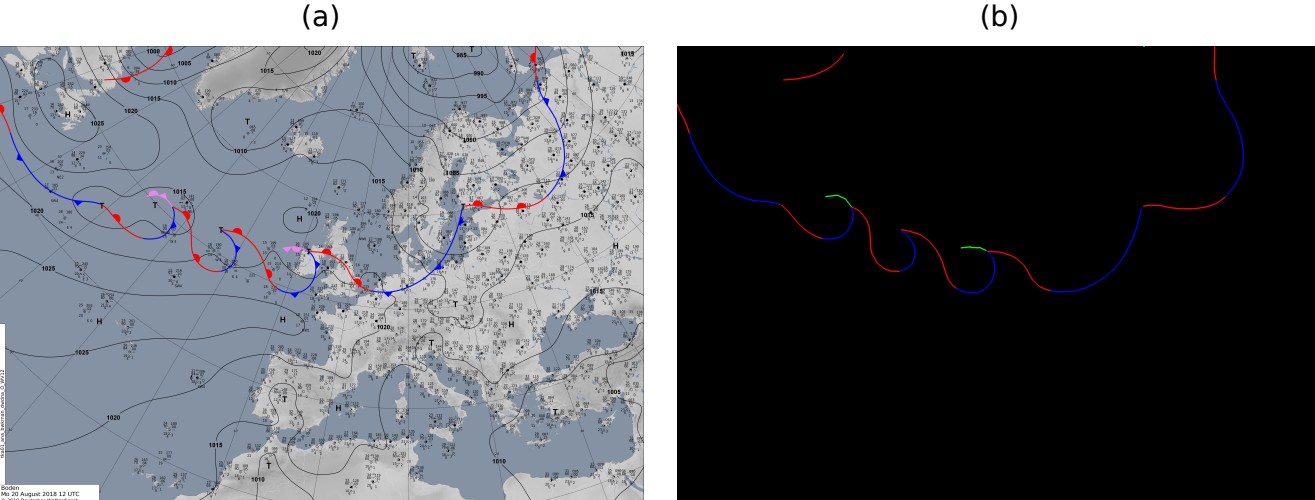

**Figure 2.** Example of well extracted fronts (b) from an image as provided by the DWD (a). Blue and red lines correspond to cold and warm fronts respectively as in the original images. Green lines correspond to the occlusions which are pink in the input image. Note that stationary fronts are originally depicted as alternating warm and cold fronts. For this reason we cannot distinguish those from regular cold and warm fronts.

our extraction method wrongly extracts these objects as well. The last cause of error occurs when we try to sort the extracted coordinate pairs of a single front. In some cases the sorting method may end up stuck in a local minimum, resulting in a wrong order of points. An example of such a faulty extracted image is shown in Fig. 3. However, these are relatively rare, only account for a small portion of fronts within a sample and many are going to be masked by the lower resolution of ERA5, which is why we ultimately decided to ignore these cases for this work.

We can extract information for the following front types: warm, cold, and occlusion fronts, respectively. Since stationary fronts are indicated by alternating warm and cold fronts, we cannot extract this information from the images as obtained from DWD; this would interfere with the classification of warm and cold fronts.

## 2.2 Network Design and Training

### 2.2.1 Network Architecture

Neural networks are a machine learning technique where a network consisting of several layers is used to extract feature representations of an input at different levels. Each layer transforms its input into an output map, the layers feature map. These feature maps can then be used as an input for consecutive layers which enables the network to learn more detailed features within the data. In a Convolutional Neural Network (CNN) the most common transformation function is a convolution of the input image with a convolution mask where each entry is a trainable, latent parameter of the network. During training these parameters are adjusted to optimize a loss function, which measures the quality of the output of the network. In our case

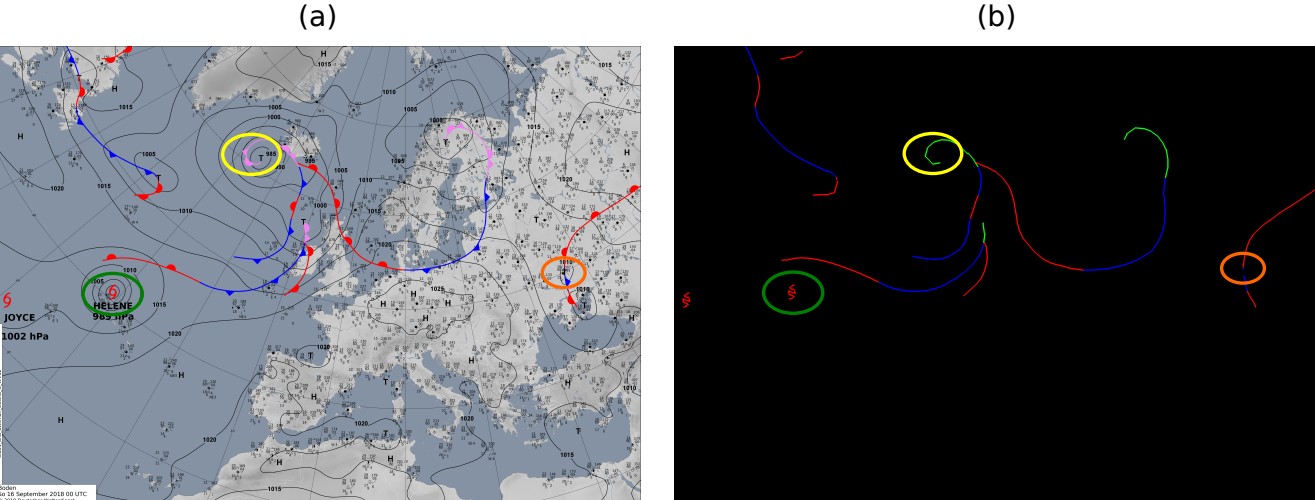

| (a) | (b) |

**Figure 3.** Example of badly extracted fronts (b) from an image as provided by the DWD (a). **Green Circle:** Object that is not a front, but has the same color coding is wrongly extracted as a front. **Orange Circle:** Unrelated symbol is drawn over the front. The front could not be extracted completely. **Yellow Circle:** Frontal symbol is placed in an area with high curvature. The curvature is not extracted exactly, as the symbol is removed during the procedure and the lose ends are connected with a straight line.

we use a U-Net Architecture originally introduced by Ronneberger et al. (2015) for biomedical segmentation. The proposed architecture consists of several consecutive blocks that gradually extract features from the data and reduce the spatial dimension of the input data to extract features on multiple scales. These blocks are followed by a number of expansive blocks which gradually increase the resolution up to the original scale. Additionally at each resolution scale a so called skip connection allows the final feature map of a encoding block to directly serve as additional input to the corresponding decoding block, displayed as grey arrows in Fig. 4. These skips improve the networks ability to localize the features, as the upsampled features only hold coarse localization information. In our network we use convolutional layers as explained before. Additionally we use Rectified Linear Unit (ReLU), Batch Normalization, Pooling, upsampling and 2D-DropOut layers, whose functionality we will briefly explain. The dropout chance at each 2D-DropOut layer is set to $0.2$.

- ReLU layers are used to introduce non linearity into the network. They transform each input $x$ as $\text{ReLU}(x) = \max(0, x)$

- Batch Normalization layers normalize the batched input to a mean of 0 and variance of 1. They can have additional learnable affine parameter.

- Pooling layer transform several input grid points to a single output gridpoint. Common operations are averagePooling or maxPooling where the grid points are combined calculating the average or maximum of the input, respectively. This operation is used to reduce the resolution of the feature map.

- Upsample layers are a simple upsampling of a grid point to increase the resolution of the feature map.

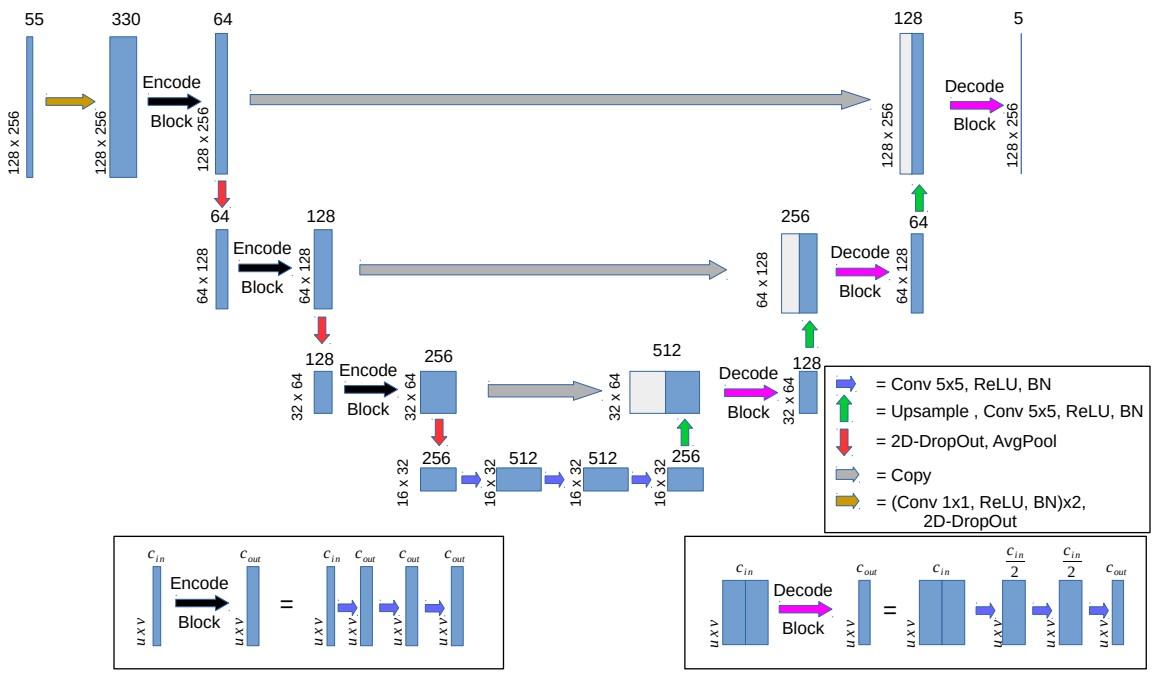

**Figure 4.** U-Net architecture used for this paper. The first convolution of the input data uses a $1 \times 1$ sized kernel instead of $5 \times 5$. Decode and Encode blocks are explained in the boxes at the bottom of the image. Each Decode and Encode block consist of 3 sequential convolution, ReLU and BN blocks. $U \times V$ describes the image size per channel. $C_{in}$ and $C_{out}$ respectively describe the number of channels of the input and output of an encode or decode block. The copy operation simply copies the blue box at the start of the arrow into the white box at the end. The white and blue boxes then describe the concatenation of the output from the copy and upsample operations. The number at the left hand side of each block denotes the spatial input dimension. The shown sizes are those used during training, however the initial spatial dimension can be chosen freely as long as it is divisible by 8. At each red (green) arrow the dimension is divided (multiplied) by 2. The number on top of each block denotes the number of channels for each block and must not be changed.

– 2D-Dropout layers randomly set all values in a channel to $0$ to reduce overfitting.

A sketch of the used architecture is shown in Fig. 4. We use Pytorch's DistributedParallel package to enable training on multiple GPUs in parallel. Training is performed on a single node, with each GPU acting on a fixed shard of the available data.

### 2.2.2 Data-Set Augmentation

In each epoch and for each timestamp we randomly select one of the available weather service labels for the given timestamp. Depending on which weather service was chosen we crop a $128 \times 256$ pixel sized sub-grid residing within the corresponding

weather services input region (see Table 2) from the ERA5 data. We use this smaller crop instead of the complete region to
increase the number of training samples, reduce the memory footprint on the GPU during training and to ensure that all input
dimensions are multiples of 8. The extracted label data is also cropped by removing each vertex, where neither the vertex itself
nor a neighboring vertex is located within the extent of the ERA5 crop. To further increase sample count via data augmentation
we also perform random horizontal and vertical flips on the data. It is important to note that, whenever data is horizontally
(vertically) flipped the sign of the input variable $v$ ($u$) has to be flipped as well, as these variables describe a vector field rather
than a stationary value. Flipping of the data might also lead to a better representation of fronts in the Southern Hemisphere,
which are "mirrored" at the equator (see video supplement Niebler (2021)).

### 2.2.3 Training

Our model is trained using stochastic gradient descent with Nesterov momentum of $0.9$ to minimize the loss function. The
initial learning rate is set to $0.005 \cdot \#Ranks$, where $\#Ranks$ corresponds to the number of processes used for the parallel
training. We train the network for several epochs. Within each epoch the algorithms randomly trains on a permutation of the
complete training data set. Every 10 epochs we measure the training loss. If the test loss does not improve for 10 test phases we
divide the learning rate by 10 up to a minimum of $10^{-7}$ and reset the count, if the learning rate was changed. If the test loss does
not improve for 20 test phases (200 epochs) and we cannot reduce the learning rate anymore we stop training. Additionally
we set a maximum of 10000 training epochs or 3 days time as stopping criteria. At each test step, we save a snapshot of the
network if the test loss is better than the currently best test loss. Our final network is the resulting network which yielded the
lowest test error.

### 2.2.4 Label Extraction

As described by Lagerquist et al. (2019) the frontal polylines are subject to two non-negligible causes of bias: inter- and intra-
meteorologist. The first bias describes the effect that two meteorologists may disagree on the exact location of a front, the
occurrence of a front at all, or which exact shape the frontal curve follows. The second bias describes the effect that the same
meteorologist may be biased on the placement of frontal data coming from previously placed fronts by the same person. The
transformation of these curves into poly-lines and the application onto a different resolution is subject to creating additional
label displacements. While these problems are present in most human labeled data it is more peculiar in this specific case
because the ideal poly-line should have a width of only *a single pixel*. As a result each ever so slight displacement introduces
a large per pixel disparity between two fronts, as the intersection of the sets of pixels that describe these fronts ends up being
close to non existent. This has at least two negative effects. First, the gradient information is really sparse, as a close prediction
will be considered false positive just as a far off prediction, as can be seen in the example of Fig. 5 a. Further translating the
green line to the right, will barely affect the count of intersecting pixel with the red line, even though one would consider the
detection becoming worse the further it moves from the label. Secondly, the previously mentioned label offset due to personal
bias may lead to the case that a labeled front is not located exactly at the physical frontal position, essentially creating a false

label with wrong underlying atmospheric properties. Due to the low intersection count, a correctly placed detection will now score badly.

One way to handle this might be to widen the extracted front label. While this approach introduces further false positive labels slight translations in the detection are less penalized as they are more likely to be covered due to the larger width of the labeled data. Additionally the network is inclined to also detect wider frontal lines, making it even easier to create intersections. In the same way the effect of positional bias of the label placement is also reduced as the widened label is more likely to cover the physically correct location, if a small translational bias exists. However, this bias is not completely negated. From our studies and the results of previous studies (e.g., Matsuoka et al., 2019; Lagerquist et al., 2019; Biard and Kunkel, 2019) it seems apparent that a deep learning architecture learns that a bias in label placement exists and as a result tends to predict enlarged lines, trying to cover the uncertainty caused by the bias. Using enlarged labels further enhances this effect, leading to even larger detections, which in return leads to a low spatial accuracy of the detections. To regain positional accuracy previous work used a morphological post-processing step to extract thin lines from wider network predictions.

In this work we use a different approach, as displayed in Fig. 5, panels b and c, to counteract this initial loss of positional accuracy. Instead of widening the label, we deform the given polylines prior to evaluation, by translating the vertices within a restricted search radius (panel b). All possible deformations are considered and evaluated according to a matching function and the highest scoring deformation is then used for evaluation (panel c). This approach encourages the network to predict fronts with a high spatial certainty, as the labels themselves remain thin, while the deformation models the positional bias.

A polyline $j$ consists of as series $v_j$ of vertices $v_{j,i}$, where each $v_{j,i}$ describes the coordinate pair of the vertex as it is extracted from the weather service label. Additionally each deformed polyline contains a series of translations $tr_j$, consisting of a translation vector $tr_{j,i} = (u_{j,i}, w_{j,i})$, which describes the translation of $v_{j,i}$ within the polyline $j$. A segment of the deformed polyline $j$ is then edge $e_{j,i}$ connecting $v_{j,i} + tr_{j,i}$ and $v_{j,i+1} + tr_{j,i+1}$. We calculate the matching score of a segment as follows.

- calculate the positions of pixel of the line connecting $v_{j,i} + tr_{j,i}$ and $v_{j,i+1} + tr_{j,i+1}$

- sum the values of all pixel in the network output that are on this line.

- weight the sum by $1 + \exp(-0.5((\frac{u_{j,i+1}}{\sigma})^2 + (\frac{w_{j,i+1}}{\sigma})^2))$

- reduce the result by the number of pixel in the line connecting $v_{j,i}$ and $v_{j,i+1}$

The matching score of a polyline is considered the sum of the matching scores of each line segment of the deformed polyline.

The third step models a prior belief that the provided labels are generally placed correctly and that strong deformations are less likely. Therefore a low deformation is preferred to a strong deformation if the intersection with the network output is the same. This matching procedure operates ignorant of the classification results and only takes the presence or absence of any type of front at a given pixel into account. We restricted ourselves to deformations where $-k \leq u_{j,i}, w_{j,i} \leq k$ with $k = 3$, keeping the deformation radius small to only counteract the positional bias of the label, which we expect to be small. Additionally we chose $\sigma = k$. We do not change classification information of the labels during the procedure. Thus each front is extracted as

the class provided by the weather service. This matching procedure was implemented using C++ and Pybind11 (Jakob et al., 2017).

This method comes at the risk, that instead of predicting the position of the front the network may end up detecting a systematic displacement of the front within the range of the $(2k+1) \times (2k+1)$ grid. We believe this could happen for two possible reasons: (i) the label bias exhibits a systematic displacement itself, and (ii) $k$ is chosen too large. In the first case the error lies within the labels and it is generally questionable whether or not these labels are suitable for training at all. The parameter $k$ controls at which distance from the labeled front the detection may still be considered correct. With increasing

$k$ the incentive to place the detection close to the provided label reduces, diminishing the spatial accuracy of the predictions. Therefore we have chosen $k = 3$, allowing each vertex to displace itself up to 3 pixel in each direction, limiting the scope of movement to a sensible range.

    As an example Fig. 5 shows how this algorithm can help to solve the problem of a correct detection being penalized by a biased label. We assume that the green line (Detection 1) is a correct detection, with appropriate underlying atmospheric

properties, while the yellow line (Detection 2) is an artifact caused by unfinished training of the network. Additionally the red line was drawn biased and is therefore not located at the appropriate position, regarding the underlying atmospheric features. In panel (a) the correct prediction has very few pixel intersecting with the label, similar to the wrong prediction. Not performing any deformation would wrongly count several pixel of the green detection as false positives while only resulting in a similarly low number of pixels considered true positive as the yellow detection. However after the deformation algorithm most pixel

of the green detection correctly counted as true positives, while the yellow detection is correctly classified as false positive. A deformation towards Detection 2 does not occur in this example, as the yellow line is out of range for most vertices. Most segments will therefore not intersect with the yellow line leading to generally lower matching scores than the displayed blue line. The latter further displays the importance of the choice of $k$, to prevent the label from deforming onto a wrong detection.

### 2.2.5   Loss Functions

During training we extract the label lines as described in Section 2.2.4. As a loss function we decided to use a loss based on *Intersection over Union* (IoU), which we evaluate for each output channel individually, before combining them by a weighted average. This loss inherently circumvents the problem that in each channel most of our output, belongs to the background as it does not contain a front. While the original formulation of IoU is used for sets and therefore a strictly binary labeling, we used an adjusted version that works with floating point probabilities. This loss function is also used by Matsuoka et al.

(2019). However, they only evaluate it on a single output channel. The definition of loss for a single output channel is shown in Equation 1:

$$L(p,x) = 1 - \frac{\displaystyle\sum_i p_i \cdot x_i}{\displaystyle\sum_i p_i \cdot p_i + \sum_i x_i \cdot x_i - \sum_i p_i \cdot x_i} \tag{1}$$

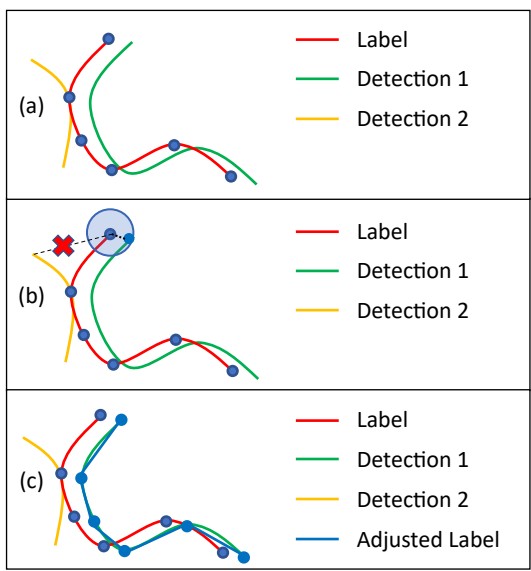

**Figure 5.** Sketch of our label adjustment method. (a) Initial Weather Service label with poly line vertices (blue dots) and 2 possible detections. Detection 1 initially scores lower than Detection 2 due to a lower intersection with Label. (b) Display of how a vertex of Label might be adjusted within a search radius for Detection 1. The possibly optimal position for the vertex regarding Detection 2 is not within the search radius of the vertex. Deformation will therefore not be able to create a good intersection of the upper part of Detection 2 and Label. A similar situation occurs for the three vertices at the bottom right of Label. (c) Possible resulting Adjusted Label after each Vertex was adjusted. The Label was deformed onto Detection 1 as it creates the best matching score. Detection 2 is too far from several vertices of Label and cannot score a similar matching score with any deformation of Label. As a result Detection 1 now scores higher than Detection 2.

Here $L$ denotes the loss function, $x$ is the extracted label image and $p$ the prediction of our network. $p_i$ and $x_i$ are respectively the $i^{th}$ pixel of either $p$ or $x$. We subtract the loss function from 1, as we will minimize our loss function during training, as the

IoU normally increases the better the prediction becomes. $L(p,0)$ evaluates always to 1 regardless of $p$, which means we do not obtain much information from such an label. When combining our networks output channels, we try to adjust for this problem. We define a variant of $L$, denoted as $L^0$, that simply omits evaluation for all $L^0(p,0)$ by setting the result to 0. In all other cases $L^0 = L$. These omitted cases therefore will not influence the training gradient. As our network generates a multichannel output we calculate a loss for each channel individually and combine the results. The first output channel corresponds to the

background label, which corresponds to the absence of fronts. We invert this output, by subtracting it from 1, to get a value describing the presence of fronts. As a result we obtain 5 output channels describing fronts (front, warm, cold, occlusion, stationary) denoted as $k \in 0, 1, 2, 3, 4$. Additionally in each batch $b$ we have $batchsize$ samples $b_n$ and for each $b_n$ we have a detection $p_{b_n}$ and a label $x_{b_n}$. The respective data in the channel $k$ is then denoted as $p_{b_n,k}$ and $x_{b_n,k}$. For each $b_n$ we calculate $L_{b_n,0} = L(p_{b_n,0}, x_{b_n,0})$. For the classification channels $k > 0$, we calculate $L^0(p_{b_n,k}, x_{b_n,k})$ instead and denote these results as

$L^0_{b_n,k}$ correspondingly. By doing so, we may omit some samples where no label is present, within the respective channels. To compensate we define a weight $s_{b,k} = \frac{batchsize}{nz_{b,k}}$ for $k > 0$, where $nz_{b,k}$ is the number of samples in $b$ where there is no label in

channel $k$. This weight is used, to balance the potentially different counts of labels for the individual channels. The resulting loss for one $b_n \in b$ is displayed in Eq. 2. The values $0.2$ and $0.8$ are chosen to formulate a weighted average over all channels. In the case of $nz_{b,k} = 0$ we set $s_{b,k} L_{b_n,k}^0 = 0$. In this case channel $k$ will not evaluated at all within the current batch. The loss

for the complete batch can then be calculated as the mean of all $E_{b_n}$ within the batch $b$ as shown in Eq. 3

$$E_{b_n} = 0.2\, L_{b_n,k} + 0.8\, \frac{\sum\limits_{k=1}^{4} s_{b,k} L_{b_n,k}^0}{4} \tag{2}$$

To obtain the per batch loss we then calculate the mean of all $E_{b_n}$ as:

$$E_b = \frac{\sum\limits_{b_n \in b} E_{b_n}}{batchsize} \tag{3}$$

## 2.3   Baseline Method

We compare our results against a baseline method provided by ETH Zurich. The method introduced by Jenkner et al. (2010) and later modified by Schemm et al. (2015) uses thermal gradients and other information to predict fronts. While the method was originally designed to work on a $1°$ resolution grid, we adjusted the hyper parameters of the method to allow it to run on a $0.5°$ grid[1]. In the baseline method, i.e. that designed for the ERA-Interim data-set with a grid spacing of $1°$, a minimum equivalent potential temperature gradient of $4 \cdot 10^{-2}\,\mathrm{K\,km^{-1}}$, a minimum advection velocity of $3\,\mathrm{m\,s^{-1}}$, and a minimum front length of

$500\,\mathrm{km}$ is used. We decided to keep these physical values identical to the original algorithm to retain similar physical properties of the front. However, we have altered parameters used for the a-priori smoothing of the equivalent potential temperature gradient field (number of filter applications as described in Jenkner et al. (2010) increased from 5 to 7), the smoothing of frontal lines (smoothing parameter changed from 5 to 15), as well as the minimum size of front objects in number of grid-points (from 15 to 20). The largest impact comes from adjusting the smoothing of the equivalent potential temperature gradient field. Using

these altered settings, the number of fronts detected in the northern and southern extra-tropics increases by about $30\,\%$, but the spatial distribution of fronts is very similar to the original ERA-Interim data-set with some exceptions in the vicinity of steep terrain (not shown). Our network works on a $0.25°$ resolution grid and outputs on the same domain. Therefore, when comparing against the baseline method we resample the network output to a $0.5°$ resolution using a 2D maximum pooling operation. The authors of the baseline method mention that the provided baseline should only be applied to the midlatitudes.

When comparing against the baseline we therefore restrict ourselves to the midlatitudes of the northern hemisphere for a fair evaluation.

## 2.4   Evaluation methods

We will briefly explain how the data is processed for the evaluation and how the Critical Success Index (CSI) is calculated.

---

[1]A tuning of the method for the $0.25°$ resolution was not possible, since features on small scales disturb the evaluation of the gradients

**Table 3.** Distribution of our data into training, validation and test data sets. For each data set the covered time frame and number of labels are shown. All models use the same validation and test data.

| Data set | years | samples |
|---|---|---|
| test data | 2016 | 1464 |
| validation data | 2017 | 1460 |
| training both | 2012-2014, 2015/03 - 2015/12, 2018, 2019 | 8526 |
| training NWS | 2012-2014, 2015/03 - 2015/12 | 5608 (only NWS label) |
| training DWD | 2015/03-2015/12, 2018, 2019 | 4142 (only DWD label) |

### 2.4.1 Trained Models and Data set distribution

We distribute our data into a test (year 2016) and a validation (year 2017) data set and create 3 training data sets as described in Tab. 3. We train a total of 3 models, one for each training set. The models trained using *training NWS* (*training DWD*) are additionally restricted to only use label data from the NWS (DWD) during training. Each model is trained using 6 GPUs on a single node of the Mogon II cluster of the Johannes Gutenberg University. Each node contains 6 Nvidia GTX1080 GPUs and an Intel Xeon CPU E5-2650 v4 with 24 cores and hyperthreading. Data was staged in prior to training to enable reading from

a local SSD rather than the parallel file system. The models trained using *training NWS* and *training DWD* are only used in section 3.1.1 with results presented in tables 4 and 5 as well as in the SI in tables S1 and S2. In all other cases the model using *training both* is applied.

### 2.4.2 Test Data processing

For the evaluation we process each input file in the test data set as follows:

– Apply the respective model on the global input region of the current sample

– Apply a softmax activation function to the raw network output to generate a probability mask for the sample.

– Create a binary mask by setting each entry in the probability mask to 1 if it is greater than $0.45$, else to $0$.

– Use one iteration of 8-connected binary dilation and calculate all different connected components. Each connected component is considered an individual front.

– Filter the labeled image with the undilated binary mask to remove the dilation effect.

– remove all fronts that consist of less than 2 pixel

– Write the binary mask to disk

During evaluation we then load the corresponding binary mask from disk and crop it to a sub-region when necessary. Results of the baseline method and the weather service labels are already provided in binary format.

### 2.4.3 Calculation of Critical Success Index (CSI)

We evaluate the detection quality of our network and the baseline method by calculating the CSI similar to Lagerquist et al. (2019). As ground truth the provided weather service label of the surface fronts is used.

**Front to object conversion:** Prior to evaluation the generated binary masks of our network output are transformed into front-objects in two steps.

- Use one iteration of 8-connected binary dilation and calculate all different connected components. Each connected component is considered an individual front.

- Filter the labeled image with the undilated binary mask to remove the dilation effect.

The same transformation is applied to the provided weather service fronts. Note that some provided weather service fronts are separate lines in the label file, but end up as a single longer front due to being connected due to the coarser grid used in the analysis, e.g. $0.25°$.

**Front-Object matching:** A predicted front $F_p$ is considered to be matched to the weather service label if the median distance of each pixel of $F_p$ to the nearest labeled pixel of the same class in the weather services label image is less than a detection radius of $D$. The same is applied vice versa for the weather service fronts compared against the network output. Each class of front can only be matched to pixel of the same class, however each frontal object is matched against the whole set of pixels of the same class, rather than just a single other object.

For the evaluation we define two distinct regions, namely (i) the evaluation region, which is the region out of which we take the fronts, we want to match against any other fronts, and (ii) the comparison region, which is the region in which the algorithm checks for possible matches for the fronts within the evaluation region. In our evaluation the comparison region is the same as the evaluation region, with an additional extension of $10°$ in each direction. The advantage of looking for matches within this comparison region instead of the evaluation region, is to reduce false results caused by the crop of the evaluation region. E.g. fronts at the edge of the evaluation region, may be split into multiple fronts due to the crop, skewing the count of individual fronts. Alternatively a front located at the edge of the evaluation region may be counted as unmatched, because the possible match was cropped out. Using the comparison region we will resolve most of these cases. A sketch of this is shown in Fig. S1. Note that using this larger region for the matching purposes does not add any fronts to the evaluation nor does it affect the matching radius $D$. This change only allows each front to better use its search radius $D$ to find possible matches, unaffected by input crop.

**Critical Succes Index calculation** We define $n_{MWS}$ as the count of fronts provided by a weather service, that could be matched against the prediction, while $n_{WS}$ is the count of all provided fronts. Similarly, $n_{MD}$ describes the count of all detected fronts, that could be matched against the weather service fronts, while $n_D$ describes the total count of detected fronts. With these values we can then calculate the *Critical Success Index* (CSI), *Probability of Object Detection* (POD), and *Success Rate* (SR) as described in Eq. 4, 5, and 6, respectively. As mentioned by Lagerquist et al. (2019) these measurements are also applied in other scenarios, like the verification of tornado warnings by the NWS (Brooks, 2004). The SR describes

the probability that a predicted front corresponds to an actual front from the labeled data-set, while the POD describes the probability that an actual front is detected by the network. SR and POD could easily be maximized at the cost of the other, by either not predicting anything or classifying each pixel as a front instead. The CSI serves as a measurement that penalizes such degenerate optimizations as it maximizes only when both values yield good results. Generally speaking a high CSI score is preferable. Whether it is more important to have a high POD or SR depends on the task at hand and whether it is more important that the detection is more sensitive or more accurate.

$$POD = \frac{n_{MWS}}{n_{WS}} \tag{4}$$

$$SR = \frac{n_{MD}}{n_D} \tag{5}$$

$$CSI = \frac{1}{\frac{1}{POD} + \frac{1}{SR} - 1} \tag{6}$$

## 3 Results and Discussion

In this section we evaluate the CSI of our network against the weather services and compare it against the baseline method. We additionally create climatologies for both methods and calculate the pearson correlation against climatologies created from the weather service data. In a second section we present further results of our networks output where we look into physical quantities across the frontal surface and the relation fronts to extreme precipitation events to infer physical plausibility of our networks detections and highlight possible scientific application scenarios for the presented method.

### 3.1 Performance Evaluation and Comparison against Baseline

### 3.1.1 Front Detection Quality

In Fig. 6 we provide an image showing an example of the networks output compared to the label of the corresponding weather service. The image shows that the network tends to create thin fronts, as desired. The detections also appear to have a generally smoother shape compared to the weather service labels. The general shape of the fronts appears plausible, even though there are disagreements between the detections and labels regarding both the shape and class of fronts. For a better image of the networks output we also provide a video supplement showing the network output on a global scale Niebler (2021). Further details are listed in Section S4 in the SI.

To quantify the quality of our predictions we evaluate the CSI, POD and SR for a matching radius of $D = 250 \, \mathrm{km}$ on our test data set and display the results in Table 4 and 5 for the binary task which only considers the classes front and no-front, as well as the individual scores for each of the four frontal classes. As evaluation region we use the corresponding weather services output region as defined in Tab. 2.

The provided results show that the network excels at the pure front detection task with CSI scores of $66.9\%$ (DWD) or $68.3\%$ (NWS). At the same time the network evaluates with a POD and SR exceeding $77.3\%$. POD tends to be higher than SR for the NWS data, while on the DWD data SR tends to be higher than POD. Overall the classification scores are comparably

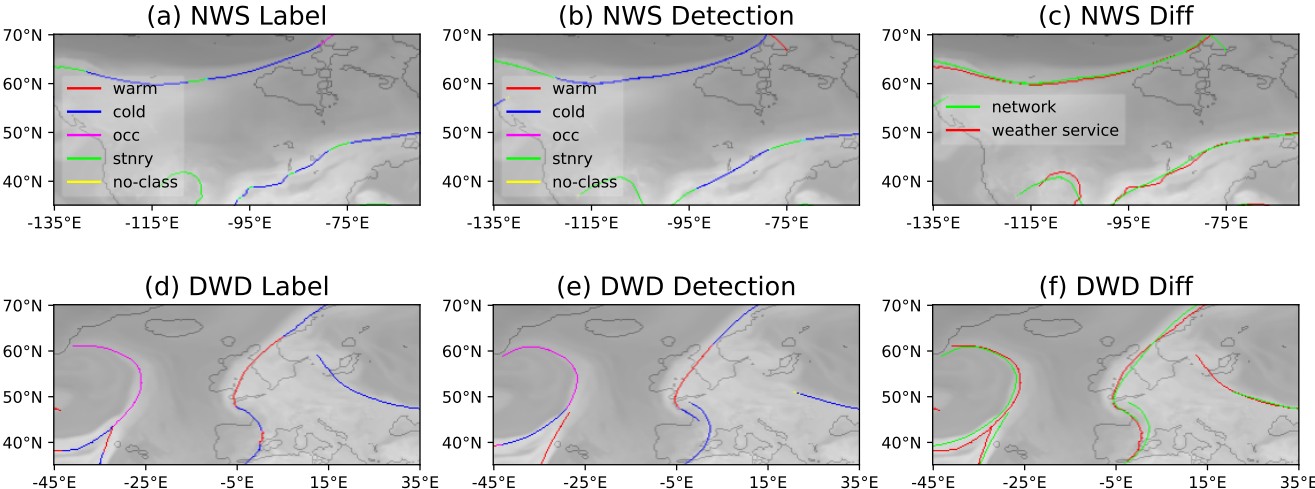

**Figure 6.** Fronts from provided labels of the NWS (a) and DWD (d) as well as the corresponding network generated outputs ((b) and (e) respectively) displayed on top of equivalent potential temperature. Colors indicate the frontal type, whereas unclassified fronts are displayed yellow. The labels are the same for both rows. The difference images (c, NWS) and (f, DWD) show a direct comparison of frontal placement by the weather service (red) and the network (green), ignoring classification. All displayed examples are at 14 September 2016, 00:00:00 UTC.

lower with a class CSI ranging between $36.4\%$ and $56.8\%$. Across all tests warm and stationary fronts appear to be harder to classify for the network than cold fronts or occlusions. This effect is more pronounced on the NWS dataset. A possible

explanation for this is the lack of a clear distinction of these two front classes from the DWD data, which in return leads to more false classifications due to the ambiguity. We can further see that training on a single region does not provide a good generalization onto the other region, which is expressed by a lower CSI scores when training on only the DWD (NWS) data and evaluating on the respective other, i.e., NWS (DWD) data. At the same time training on both regions yields comparable scores as the single region trained networks. This clearly shows that using the method trained on both regions is preferable. We

will therefore continue our evaluation with only this model. This difference between the regions may be originating in different synoptic structures of cyclones and their associated fronts over the North American continent and over the North Atlantic. This implies that the inclusion of further data-sets - for example the data-sets used by Matsuoka et al. (2019) or generally data of the southern hemisphere, may improve the networks performance even further. This is also interesting regarding a thorough evaluation of the networks performance on the southern hemisphere. We want to point out here that the inclusion of additional

training data of similar structure than the used NWS/DWD data can be carried out easily, the method is designed to be very flexible.

  We also evaluated results where each object can only be matched against a single object of the corresponding class instead of the whole set. These are listed in Tables S1 and S2. We observe a drop in POD from 77.3 (83.4) to 70.8 (76.9) when evaluating on DWD (NWS) data, while SR barely changes. This indicates that our network tends to not fully cover large frontal regions

with a single front but rather multiple smaller, disjointed fronts. Each of these can still be matched against the large front but none of them is sufficiently large enough such that the large front can be matched against any one of them, leading to the lower object detection rate. Interestingly, we also do not observe the same change in POD when only considering the classification scores. This further indicates that the previously mentioned fragmentation does not occur within the individual classes but rather at the transition between classes. When the weather service labels several fronts of different classes as connected, the

generation of the binary label merges all these fronts into a single long front. If the network then is able to detect the individual fronts, but does not detect them as connected, the conversion to the binary detection will result in several shorter fragments instead. A similar effect may occur if some parts of the long front are simply not detected at all. However, the low change in the classification scores indicates that the first effect is more pronounced. In the bottom row of Fig. 6 an example of such a fragmentation can be seen, where the network detects the central front as two separate fronts, while the provided label is a

single connected front. Using the initially introduced matching method, where each front can be matched against the whole set of a class the fragmentation problem can be overcome. At the same time SR and classification scores are barely affected which shows that this method is suitable for our task.

**Comparison against Baseline:** We additionally evaluated the CSI score on a coarser $0.5°$ resolution grid and compare the results against the baseline algorithm, evaluated on the same grid. The used baseline does not classify its results which is why

we only display and compare the task of front detection and forgo any classification results. Due to the previously mentioned fragmentation issues, we only evaluate the results where each front may be matched against the complete set of fronts rather than just a single front object. The baseline algorithm is only designed for the application in the midlatitudes and should not detect stationary fronts. Hence for this comparison we further restrict our evaluation region to fit within the midlatitudes of the northern hemisphere and remove stationary fronts from the labels and network output. There may be an offset between the

placement of a front by the baseline and the weather services as the baseline locates its fronts at the center of a passing front rather than the leading edge. While we believe that the used matching procedure already respects such a difference we also evaluated the baseline method using $D = 500$km, doubling the search radius compared to our network. As shown in Table 6 our network (NET) outperforms the baseline algorithm (baseline) in all evaluated scenarios and metrics with a more than twice as high of a CSI score when using $D = 250$km. Even when the baseline is evaluated with a larger search radius of $D = 500$km

the network outperforms it with a difference in CSI scores of more than $10\%$, even though the network is still evaluated using the smaller search radius of $D = 250$km.

### 3.1.2 Comparison of Frontal Climatologies

To further investigate the soundness of our predictions we created frontal climatologies for the year 2016 for both the provided weather service labels as well as our network and the baseline method. While the respective weather services only provide

labels within their analysis region, both the network and the baseline can be executed on the global grid. As in section 3.1.1 we explicitly remove stationary fronts from both the NWS label dataset as well as the network output, when creating those climatologies. This is done as the baseline method does not include fronts propagating at less than $3\,\mathrm{m\,s^{-1}}$. The baseline was designed for application within the midlatitudes, and results outside the midlatitudes should be taken with care. We therefore

**Table 4.** CSI, POD and SR values for $D = 250$ km evaluated on DWD data for 2016. Warm fronts tend to be detected worse than the other classes while cold fronts are generally well detected. Stationary fronts are not available for DWD labels and are therefore not listed. Evaluation regions contains latitudes within $]35°, 70°]N$.

| Training region | NWS | | | DWD | | | Both | | |
|---|---|---|---|---|---|---|---|---|---|
| | CSI | POD | SR | CSI | POD | SR | CSI | POD | SR |
| Binary | 51.1 % | 65.4 % | 70.1 % | 68.4 % | 78.7 % | 84.0 % | 66.9 % | 77.3 % | 83.2 % |
| Warm | 20.3 % | 22.8 % | 65.1 % | 49.3 % | 58.1 % | 76.6 % | 49.2 % | 57.6 % | 77.0 % |
| Cold | 39.5 % | 47.9 % | 69.2 % | 56.6 % | 67.8 % | 77.3 % | 56.1 % | 66.3 % | 78.5 % |
| Occlusion | 35.4 % | 44.0 % | 64.6 % | 51.9 % | 69.5 % | 67.3 % | 52.4 % | 67.2 % | 70.3 % |

**Table 5.** CSI, POD and SR values for $D = 250$km evaluated on the NWS data 2016. Warm fronts tend to be detected worse than the other classes while cold fronts are generally well detected. The network trained purely on DWD data, could not learn stationary fronts, as they are not included in the training data, which is why these are not listed. Evaluation regions contains latitudes within $]35°, 70°]N$.

| Training region | NWS | | | DWD | | | Both | | |
|---|---|---|---|---|---|---|---|---|---|
| | CSI | POD | SR | CSI | POD | SR | CSI | POD | SR |
| Binary | 67.3 % | 81.9 % | 79.1 % | 49.7 % | 57.0 % | 79.6 % | 68.3 % | 83.4 % | 79.1 % |
| Warm | 37.3 % | 56.5 % | 52.4 % | 22.5 % | 44.1 % | 31.6 % | 36.4 % | 58.1 % | 49.3 % |
| Cold | 55.6 % | 70.1 % | 73.0 % | 41.2 % | 51.8 % | 66.8 % | 56.8 % | 73.1 % | 71.8 % |
| Occlusion | 48.7 % | 72.5 % | 59.8 % | 36.1 % | 62.7 % | 46.0 % | 49.0 % | 73.4 % | 59.5 % |
| Stationary | 44.6 % | 59.4 % | 64.1 % | | — | | 43.2 % | 56.2 % | 65.2 % |

restrict our quantitative evaluation to regions within the midlatitudes. We nonetheless present the climatology on the global area to emphasize the difference in performance of the network compared to the baseline outside the midlatitudes. The resulting climatologies are shown in Fig. 7.

First we compare the climatology for the North Atlantic / European region from the manually labeled data-set with the climatology of network generated fronts. In the DWD climatology the North Atlantic storm track is clearly visible as a band of heightened front occurrence stretching from the East coast of North America to the channel (Fig. 7 c). Frontal activity is tampering off inwards of the European west coast. The climatology of the network generated fronts has a very similar overall structure with a strongly enhanced frontal frequency in the storm track region (Fig. 7 a). Frontal frequency is somewhat larger at the beginning of the storm track. This may be related to the training with North American manual analysis, which naturally has a stronger focus on the early cyclone lifecycle than the European data. Over the Channel and North Sea Coast of Europe frontal frequency in the network generated data-set is somewhat lower than in the DWD data-set, which may be related to the inclusion of stationary fronts in the latter but not the former. We have seen also in the previous section that very weak warm fronts, as may exist further into the European continent are often not detected by the network. In both data-sets a slightly

**Table 6.** Comparison of the CSI, POD and SR of the baseline algorithm against our network for the data of 2016, restricted to the midlatitudes in the northern hemisphere. As the algorithm provided by the ETH does not classify fronts we use the binary-classification evaluation for our network. (quasi-)stationary fronts were removed from the network output as well as the NWS label, as the baseline Algorithm should not predict those. For the DWD Label these could not be reliably removed, due to the labels ambiguity. We can see that the baseline algorithm is better in predicting fronts in the DWD regions rather than the NWS region. Evaluation performed at $D = 250$km for NET and baseline$_{250}$, evaluation performed at $D = 500$km for baseline$_{500}$. However, the network performs better in terms of all three measures for both regions.

| Method | Evaluation on DWD Region | | | Evaluation on NWS Region | | |
|---|---|---|---|---|---|---|
| | CSI | POD | SR | CSI | POD | SR |
| baseline$_{250}$ | 31.2 % | 44.4 % | 51.2 % | 21.9 % | 42.7 % | 31.1 % |
| baseline$_{500}$ | 56.4 % | 68.0 % | 76.6 % | 48.1 % | 69.9 % | 60.7 % |
| NET | 69.9 % | 78.0 % | 87.1 % | 60.2 % | 78.8 % | 71.8% |

enhanced frontal frequency around Iceland is evident.

Next we compare the climatology for the North American region from the manually labeled data-set with the climatology of network generated fronts. The manual labels indicate the onset of the storm track with enhanced frontal frequencies just off the
North American East Coast and secondary peaks in frontal frequencies in the lee of the Rocky Mountains and along the West Coast (Fig. 7 d). The climatology of network generated fronts captures all three maxima in the frontal frequency in roughly the same location (Fig. 7 a). However, frontal frequency in the lee of the Rocky mountains and along the West Coast are more pronounced in the network generated climatology. We are under the impression that the network tends to assign labeled warm fronts as stationary and vice versa. These shifts may explain the different frontal frequency.

Finally, we compare the global climatology of network generated front labels to those generated by baseline automatic front detection algorithm (compare Fig. 7 a and b). The striking first difference between the two climatologies is the much larger spatial extend of regions with high frontal frequency in the second data-set. This is evident both in the storm track regions on both hemispheres but also the subtropical regions. In the subtropics regions of large gradients in equivalent potential temperature exist and these are picked up by the automatic front detection algorithm. However, their structure and origin differs from fronts in the extratropics. It appears that the network is able to detect this difference in the structure, while focusing solely on equivalent potential temperature and frontal propagation speed is not enough information to find these structures. In absence of any manual data-set that can serve as ground truth it is difficult to judge the physical meaningfulness of the climatological patterns emerging from either algorithm. And indeed in the case of the subtropics may strongly depend on the purpose and definition of what is considered a frontal structure. In the storm track regions on both hemispheres both data-sets show consistently enhanced frontal frequencies over similar geographical regions. They only differ in the zonal extend of the regions with enhanced activity and the absolute values of frontal frequencies. In the only region, where we have an independent, manually generated data-set often considered as the "ground truth", the climatology of network generated fronts is in closer agreement with the former than the climatology from the baseline automatic front detection algorithm. For the southern

**Table 7.** Extent of the regions used during comparison of climatologies. These regions correspond to the output regions used during training, limited to $[35°N, 60°N]$.

| Weather Service | Latitudes | Longitudes |
|:---:|:---:|:---:|
| DWD | $]35°N, 60°N]$ | $[-45°E, 35°E[$ |
| NWS | $]35°N, 60°N]$ | $[-135°E, -60°E[$ |

hemisphere or the North Pacific we currently do not have any such data-set available. The second striking difference is the high frontal frequency along orographic barriers in the climatology from the baseline automatic front detection algorithm, i.e. along the Andes, Greenland, Himalaya and Antarctic coast line. These maxima in frontal activity are largely absent from the climatology of network generated fronts consistent with the manually labeled data-sets. It appears that the network correctly discriminates between temperature and humidity gradients arising only because of the presence of significant topography from those caused by dynamically generated air mass boundaries. In contrast, focusing solely on the advection speeds in regions of large equivalent potential temperature gradients seem not to suffice. Overall, the global picture emerging from the extrapolation of the network trained on the North American, North Atlantic and European domain performs well also on a global scale and correctly identifies regions of high frontal activity expected from previous investigations and the known general circulation patterns. While physically plausible, this is of course no vigorous evaluation of the performance of the extrapolation to different regions of the globe. Future work should investigate this aspect in a more quantitative manner with manually labeled data-sets from other parts of the globe. Overall, the investigation of the front climatology agrees well with physically expected patterns and climatologies from manually generated frontal data-sets. This lends additional physical credibility to the network generated frontal labels. A physically plausible global climatological pattern further suggests that the learned frontal identification can be extrapolated from the training region. We found that for this it is necessary to include data from two sufficiently different geographic regions, i.e. North America and North Atlantic / Europe, as well as to augment the data-set by including also zonally mirrored examples of the frontal cases (not shown). The latter was found to be particular important for a good performance in the southern hemisphere. This is also visible in the video supplement, where the general shape, composition and motion of fronts detected in the southern hemisphere appears plausible. At first the qualitatively good results on the southern hemisphere appear to contradict our claim in the previous section, that training on a single region is insufficient of extrapolation onto other regions. However, we believe that this is due to the fact that this region is mostly covered by sea. As a result there is far less orographic influence in the southern regions. As such the simple mirroring of data from the North Atlantic may be sufficient enough to learn a seemingly good model for the sea covered regions of the southern hemisphere. Nonetheless this is only a qualitative observation, that needs to be explicitly evaluated, if appropriate data is available.

To quantify the former qualitative discussion of the climatologies we evaluated the Pearson correlation coefficient of the created climatologies within the regions described in Table 7. The resulting correlation coefficients, provided in Table 8, show that our network outperforms the baseline algorithm in both regions, with correlation coefficients greater than $77.2\%$. For both regions the networks results are more than $10\%$ higher than those of the baseline. This effect is more pronounced on the DWD

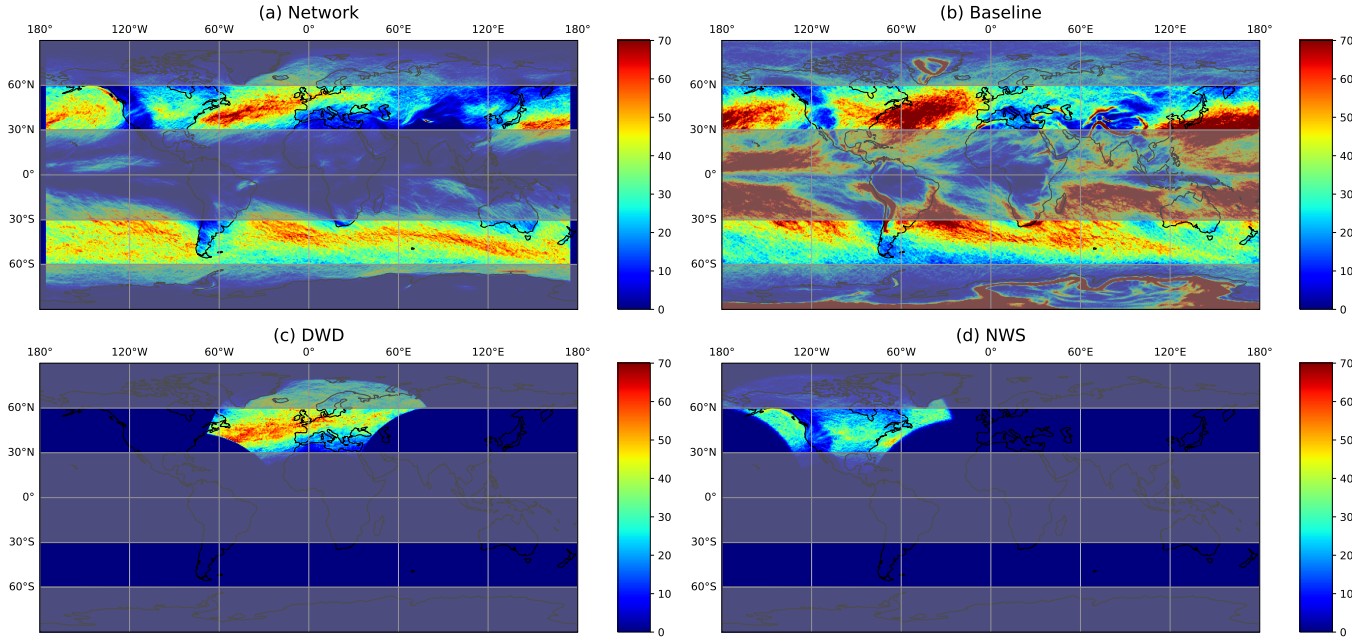

**Figure 7.** Global frontal climatologies as obtained from the ERA5 data set for the year 2016, and climatologies from the ground truth data sets of the weather services. (a) A global frontal climatology of the Network executed on the $0.25°$ resolution grid and resampled to $0.5°$ resolution. The network does not provide a valid prediction for the outer $5°$, as the effective output domain is smaller than the input domain. For this reason no fronts are displayed here. (b) A global frontal climatology of the baseline algorithm. Note that the algorithm is not designed for application outside the midlatitudes and should only be evaluated outside the gray overlayed regions. (c) Climatology of the provided DWD Front labels. (d) Climatology of the provided NWS Front labels. Red denotes more than 70 fronts. The front count was clipped at 70 for visual representation. Stationary Fronts are explicitly excluded from the climatology of the network generated data and NWS labeled data. The global climatology from the baseline algorithm does not include fronts propagating at less than $3 \, \mathrm{m \, s^{-1}}$. The DWD data-set may include stationary fronts, as we were unable to reliably separate them from warm or cold fronts.

**Table 8.** Pearson correlation coefficient of the detected climatology of the baseline Algorithm (baseline) and our trained Network (NET) against the climatologies created from the provided labels of the weather services for 2016. The columns denote the weather services, against which the methods were evaluated in the corresponding regions, limited to the midlatitudes. Stationary fronts were excluded from all climatologies except the DWD labels.

| Method | Correlation against DWD | Correlation against NWS |
|--------|-------------------------|-------------------------|
| baseline | 58.4% | 65.7% |
| NET | 79.6% | 77.2% |

dataset, which might be caused by the ambiguity of stationary fronts.

## 3.2  Evaluation of physical variables

In the previous section we showed that our proposed network can reliably detect fronts as they are provided by the weather services. In this section we will now evaluate physical properties of the detected results. We will first investigate how variables change across frontal borders and then present an application of our network evaluating, how the detected results coincide with extreme precipitation events.

### 3.2.1  Variation of Physical Variables across Frontal Surfaces

In this chapter we evaluate various physical quantities across the detected frontal borders qualitatively, to assess whether or not the detected fronts express plausible physical features. Since some automatic methods as e.g. the baseline method rely on gradients of certain thermodynamic variables, we investigate these variables for our network detected fronts. Thus, we can evaluate if these fronts are detected in a completely different way or have the same features as the frontal characteristics used for the thermodynamic methods.

We create such a cross section for each pixel that corresponds to a front in $4$ steps.

- Estimate the direction normal vector of the front at the given point

- Sample points in the normal direction centered at the given point on the front

- calculate the mean wind direction along the sampled points

- Use the sign of the dot product of the mean wind direction vector and the normal front vector to sort the sampled points
along wind direction

These cross sections are carried out at the $850\mathrm{hPa}$ level, since the TFP methods usually are based on variables on this level. For the comparison with the thermodynamic front detection methods we use the variable equivalent potential temperature ($\theta_e$). Additionally, the variables temperature, relative humidity, and (absolute) wind speed are chosen, showing important features of different front types.

The results are accumulated and the mean is presented in Figures 8 and 9 for the DWD frontal data-set. The corresponding plots for the NWS front data-set are shown in the supplement (Figures S2 and S3). In Fig. 8 (a) we evaluated the variation in equivalent potential temperature ($\theta_e$) at $850\mathrm{hPa}$ based on fronts locations (i) identified by the machine learning algorithm (dashed lines) and (ii) indicated in the surface analysis from the DWD (solid lines). For both front location data-sets $\theta_e$ is clearly increasing (decreasing) across the frontal surface for cold (warm) fronts, as would be expected from the physical definition 590 of these features. For the identified cold fronts the across-frontal temperature variation is on average larger than for the DWD labels. For warm fronts the across-frontal change in $\theta_e$ is similar for both detections, albeit the decrease ahead of the passing front is stronger for the machine learning detections, while warm fronts identified by DWD are on average located at slightly cooler temperatures. This may be explained by the assignment of some warm fronts with weak temperature gradients to the additional category of stationary fronts by our machine learning algorithm. Note that this category does not exist in the DWD

data-set. For occluded fronts there is only a small across-frontal variation in $\theta_e$ as could be expected and again this is consistent across both data-sets.

  For most automatic front detection algorithms the across-frontal $\theta_e$ gradient is of importance; this quantity is shown in Fig. 8 (b). The $\theta_e$ gradient is calculated using finite differences across the sampled temperature. Again we see very similar patterns for both the DWD and our front data-set. In both data-sets the frontal surface is located at the onset of a region with

strong change in the horizontal $\theta_e$ gradient. This is consistent with the physical definition of frontal zones and agrees with the manually designed automatic front detection algorithms. Generally the machine learning detected fronts exhibit a stronger gradient compared to the weather service for all types of front. Taking the gradient of the $\theta_e$ gradient (See Fig. 8 (c)) we obtain a size similar to the TFP, where the direction is defined by the normal of our detected front with respect to the wind direction instead of the $2D$ gradient of $\theta_e$. For simplicity we will refer to it as approximate TFP for the reminder of this paragraph.

Several traditional methods place the front at the position where the gradient of the TFP is zero. We can clearly see this for the provided DWD labels, where all 3 types of fronts have a minimum of the approximate TFP at the frontal position. For cold fronts our networks placement seems to agree with this. For stationary fronts the signal is less clear but the front also appears to be located at the extremum of the approximate TFP. Differently, warm fronts and occlusions are placed with an offset of approximately 60km to the extremum of the approximate TFP. Nonetheless we also believe that this offset is reasonable.

This shows that both our used labels but also the networks detections are plausible with respect to the theoretical background used for TFP methods. As mentioned before typically fronts are placed where the gradient of the TFP equals zero, which is thought to describe the leading edge of a front, such as it occurs with the weather service labels. The used baseline method however is different in that regard as it locates a front where the TFP equals zero, which corresponds to the center of the frontal area. This of course creates an inherent offset in the fronts position. Following this evaluation, we can see that this offset is

approximately 130km (80km) for warm (cold) fronts, both distances being lower than our used evaluation distances of 250km and 500km. This highlights that the difference in CSI should not be fully accounted to the methodological difference but rather it further enhances our statement that our network is better at the detection and placement of fronts than the baseline. In Figure 9 we additionally show the temperature in $K$ (b), relative humidity (c) and absolute wind speed (d) across the frontal border. The course of the temperature across the fronts is quite similar for both, network and weather service detected fronts, and is

physically reasonable. For instance, the temperature difference for warm and cold fronts are clearly visible; also the values agree quite well. For the relative humidity, there are some differences in the absolute values; the network detected fronts have usually enhanced relative humidity values. However, the course of the function is well captured. For warm fronts, and also occlusion fronts, there is a pronounced maximum in $RH$ ahead of the front, which indicate the typical frontal cloudiness. A similar signature can be seen for cold fronts, where the maximum is only slightly shifted as compared to the surface front

position. For the absolute wind speed we see similar values for the different fronts (detected by network vs. weather service), but no pronounced structure. Note here, that the mean absolute wind speed for stationary fronts is quite high ($|u| \sim 6-8\,\mathrm{m\,s^{-1}}$) as compared to the threshold criterion for the TFP method. However, the standard deviation is also quite high ($\sigma_u \sim 4\,\mathrm{m\,s^{-1}}$). A reason for this might be that the position of stationary fronts is not well captured by the network (also because they are only available in the NWS training data set). Due to the uncertain position, the mean values are smeared out over a large range

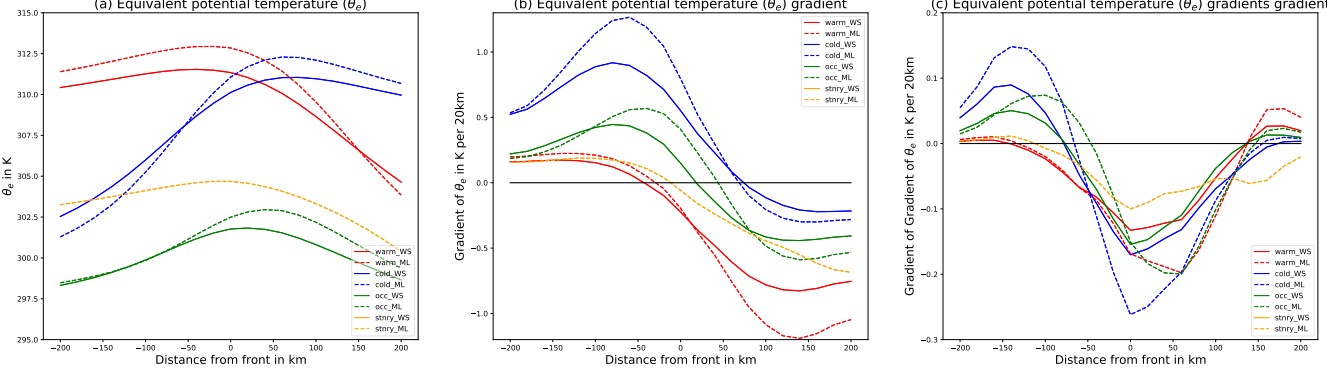

**Figure 8.** Average value of variables at $850\text{hPa}$ accross front in direction of wind.(a) Mean of the equivalent potential temperature ($\theta_e$), (b) $\theta_e$-gradient and (c) gradient of $\theta_e$-gradient of provided (solid, WS) and network generated (dashed, ML) front labels for the DWD Data. For (b) and (c) we additionally display the 0 level.

around the detected position. Nevertheless, the absolute wind speed of stationary fronts is much smaller than the speed of the others, because these fronts are moving quite slow - this feature is still well captured in the network detection.

When comparing the frontal zone structure over North America according to NWS labels and our generated labels, generally also consistent structures are found (see SI) with deviations mirroring broadly those identified for the DWD data.

Overall, from the good agreement in physical structures across the identified frontal surfaces from our algorithm and from the manual weather service analysis we can conclude that our algorithm detects physically meaningful positions. The positioning of the frontal surfaces is further consistent with physical intuition and interpretation prevalent in literature, and also with the physical constrains for the detection of fronts using a method based on thermodynamic variables.

We can finally remark that even using the surface front as a proxy for the synoptic scale phenomena front (as transition of air masses), the related structures either for the fronts manually determined by the weather services or automatically determined by our network are physically meaningful. This analysis shows that indeed we can use surface fronts as a ground truth for the detection of fronts in reanalysis data sets.

### 3.2.2 Correlation to extreme precipitation events

In the previous section we showed that our model detects fronts in concordance with physical expectations. We further showed that our method generally agrees with the theory of TFP methods, showing that our model predicts physically plausible fronts. In this chapter we will now further validate our results and at the same time provide an example how our proposed method may be applied in a scientific context outside of front detection for operational weather forecasts. To do this we evaluate how weather fronts as detected by our network are connected to extreme precipitation. Catto and Pfahl (2013) previously conducted such a study using a front detection algorithm based on Thermal Front Parameters (TFP) on the ERA-Interim data set. Due to

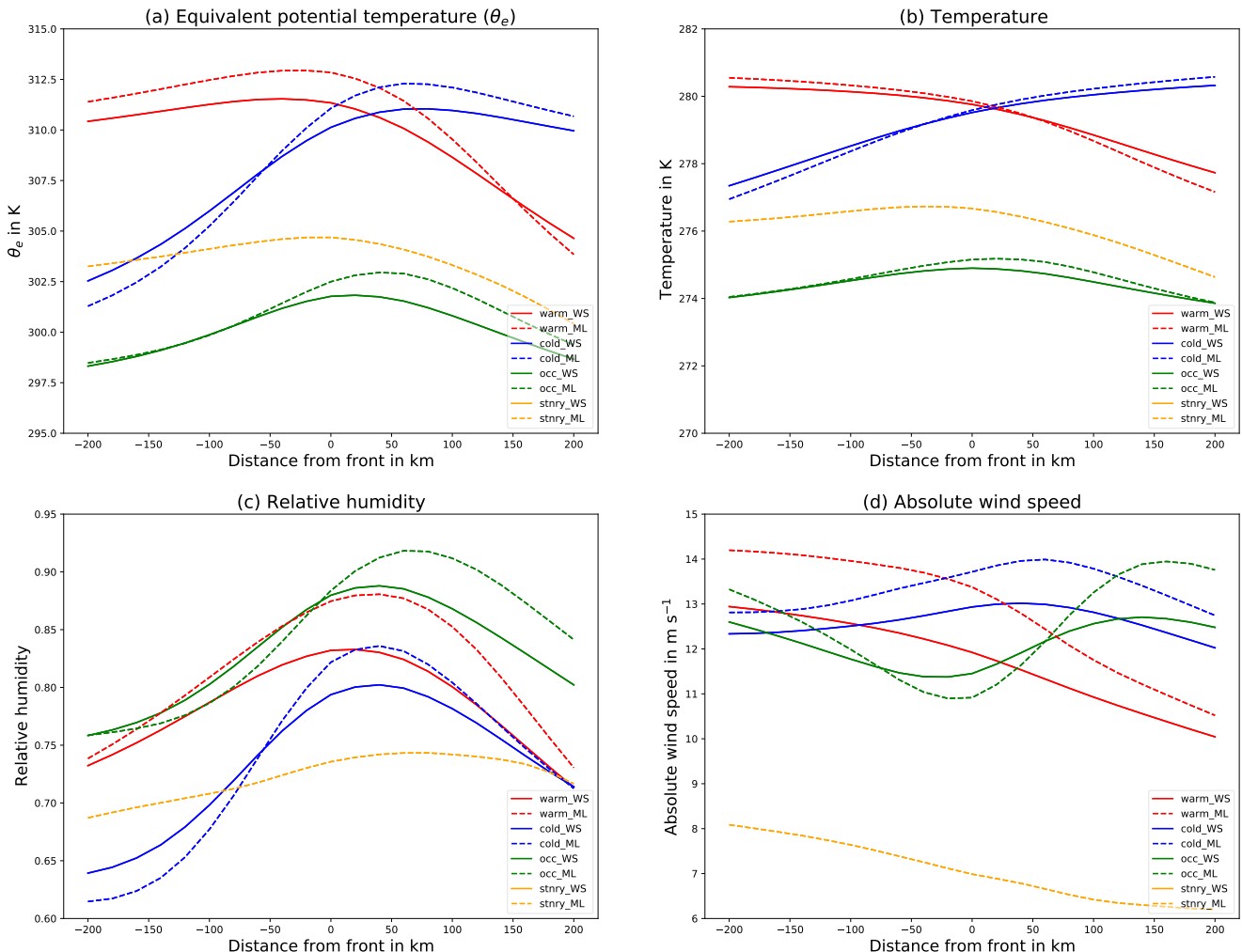

**Figure 9.** Average value of variables at $850\mathrm{hPa}$ accross front in direction of wind. (a) Mean of the equivalent potential temperature ($\theta_e$), (b) temperature, (c) relative humidity and (d) absolute wind speed of provided (solid, WS) and network generated (dashed, ML) front labels for the DWD Data.

a lower resolution of the front detection algorithm they evaluated their results on a 2.5° spatial resolution and they only use the 6 hourly accumulated precipitation variable of ERA-Interim.

**Spatial resolution:** Differently to Catto and Pfahl (2013) our front detection can be applied on the $0.25°$ resolution of the ERA5 dataset to provide a more detailed evaluation. Additionally, ERA5 provides data at an hourly interval allowing us to evaluate at a 6 times higher temporal resolution. Unlike Catto and Pfahl (2013) we decided to use the 1 hourly accumulated total precipitation to match the temporal resolution of our data samples. As all evaluation data is taken directly from the ERA5 grid we do not need to perform any resampling of data. We evaluate the data on a near global region spanning from $[-60°N, 60°N]$ and $[-175°E, 175°E]$. Grid points poleward of $60°$ are excluded as in Catto and Pfahl (2013), while the restriction within the longitudinal direction is caused by our networks reduced output domain size. We further mask high altitude regions by removing all grid points within a 5 pixel distance from any grid point exceeding a height of $2000$m from the evaluation. This filtering mostly removes stationary fronts associated with large mountainous terrain. The height of a grid point is derived from the geopotential provided in the ERA5 data.

**Data and Definitions:** For the determination of precipitation events, we use the surface precipitation as contained in the ERA5 data set (2D field for each time step). As in the study by Catto and Pfahl (2013), extreme precipitation is defined as any precipitation event that exceeds the 99th percentile of precipitation in each grid point respectively. Due to a limitation of available data we calculate this percentile using ERA5 data ranging from the years 2010 until 2018 (inclusive) using CDO. We define any grid point within an L2-distance of $2.5°$ (i.e. 10 grid points) to a front (extreme precipitation event) are considered to be associated with a front (extreme precipitation event). This a refined definition as compared to the one used by Catto and Pfahl (2013), which is possible due to the higher resolution. We evaluate the connection between fronts and extreme precipitation using $N = 8784$ samples from the year 2016; we chose this year, as it was not used during the training of our network.

We need to define a few variables for the evaluation For each grid point $p$ we define the number $N_{evt}(p)$ of different events $evt$ as the count of a $evt$ occurring at $p$ during 2016. For a grid point $p$, the counted events are as follows:

- $epr$: An extreme precipitation event occurred at $p$

- $a(epr)$: $p$ is associated with an extreme precipitation event

- $fr$: A front occurred at $p$

- $a(fr)$: $p$ is associated with a front

- $x + y$: Events $x$ and $y$ occur at the same time at $p$ (e.g. $epr + a(fr)$ describes the event that an extreme precipitation event occurs at $p$ while $p$ is associated with a front.)

.

We further define the Proportions $P_{evt}(p) = N_{evt}(p)/N(p)$ for events $evt$ as defined above. Finally we also calculate the relations

- $R_1(p) = \frac{N_{a(fr)+epr}(p)}{N_{epr}(p)}$, describing the Proportion of extreme precipitation events at grid point $p$ that can be associated with a front

- $R_2(p) = \frac{N_{fr+a(epr)}(p)}{N_{fr}(p)}$, describing the Proportion of fronts at grid point $p$ that can be associated with an extreme precipitation event.

These definitions are slightly similar to the formulation of conditional probability.

**Statistical Test:** To decide whether a connection between extreme precipitation and fronts is significant we first need to conduct a statistical test to define significance. For our investigations, we adopted the test procedure as described in the study by Pfahl and Wernli (2012).

If we assume that both events, i.e. the occurrence of extreme precipitation and a front, are completely uncorrelated, we would expect $R_1$ to be similarly distributed as $P_{a(fr)}$, i.e. the proportion of point $p$ being associated with a front. For each grid point $p$ poleward of 20°N we calculate the frontal frequency $P_{a(fr)}(p)$. We then distribute all points $p$ according to their respective frontal frequency, into bins for each $1\%$. For each bin with at least $m$ entries we randomly select $m$ grid points (base points) and create 1000 event lists, each containing $k$ successive extreme precipitation events sampled at 6 points. Each of those points is located at the respective opposite hemisphere from the corresponding base point. $k$ is chosen as 50 such that we obtain at least 300 samples of extreme precipitation events for each base point. As result for each frequency bin we obtain $m$ sampled distributions of the proportion of extreme precipitation events occurring while the base point is associated with a front. Taking the median of each of those samples we get a sample of $m$ points per bin. We then apply a percentile regression on this data to obtain linear functions describing the $1st$ and $99th$ percentile of our data with respect to the frontal frequency. We then define that for each grid point $p$ where $R_1(p)$ is not within the limits described by these percentiles respecting the underlying frontal frequency a significant connection between extreme precipitation and frontal frequency exists. We are then able to additionally mask all grid points where no significant connection could be observed.

For our test $m = 12$ was chosen as the maximum observed frontal frequency was around $53\%$. Ignoring bins with less than $m$ entries a total of 576 base points were considered. This test will be used for the evaluations in different scenarios.

**Results:** Now we present the results (i) for the occurrence of extreme precipitation if there is already a front, and (ii) for the presence of a front, if an extreme event occurs at a grind point.

**Extreme precipitation associated with fronts:** In Tables 9 and 10 the values of $R_1$ for different regions is presented. For comparison with the former work by Catto and Pfahl (2013), we report values for the global evaluation, i.e. including the tropics, although the application of front detection methods in these regions remains questionable. In addition, we present a more detailed analysis for different parts of the midlatitudes (see tab. 10). We can clearly observe that a higher proportion of extreme precipitation events can be associated with fronts when considering sea covered surface points. A similar correlation can be seen if high mountains are filtered out; again, the correlation between extreme precipitation and fronts increases. Over flat terrain, the frontal systems can develop in a quasi idealized fashion, thus warm, cold and occlusion fronts can develop quite undisturbed. Thus, extreme precipitation is mostly linked to the large scale features, whereas over (steep) terrain, local effects can disturb these developments. This effect also explains why $R_1$ is higher for the southern midlatitudes or hemisphere compared to their northern counterparts. Further we can see that $R_1$ is higher for the midlatitudes than for the tropics for all types except stationary fronts, where we observe the opposite effect. This is expected as it coincides with the frontal frequency at these locations. While stationary fronts are more often detected near high altitude regions, above land surface and at the ITCZ,

the other types of fronts tend to occur more often over the ocean, e.g. the storm tracks in Atlantic and Pacific, respectively. This is in accordance with the observations from Tabs. 9 and 10, where we can see the same connections for $R_1$. Figure 10 displays $R_1$ for each frontal type at each grid cell. For this plot all high altitude regions are grayed out (light gray), while all regions where no significant connections between fronts and extreme precipitation could be found are whited out. Further we masked all regions where no extreme precipitation event was found using a dark gray overlay. This may occur since extreme precipitation is defined using all years from 2010 to 2018 while evaluation is only performed for the year 2016. In the storm track regions over the ocean we can see regions where more than 90% of all extreme precipitation events can be associated to a front. Over all extreme precipitation appears to be more often associated with cold fronts than warm fronts. Especially at the northern midlatitudes we can see that extreme precipitation being associated with warm fronts occurs farther north than being associated with cold fronts. For occlusions this is even more clear as the highest proportion of extreme precipitation being associated with occlusions is close to $60°N$. For the southern hemisphere, a similar tendency can be seen, even though the extremes are not as clearly visible. As previously mentioned stationary fronts are less often to be found within the oceanic regions of the midlatitudes, which is why almost no extreme precipitation events are associated with stationary fronts there. Extreme events in the tropics, especially at the ITCZ are more likely to be associated with stationary fronts. Similarly the eastern parts of North America and Land surface near the north eastern pacific coast of Asia also have a relatively high percentage of extreme precipitation being associated with stationary fronts.

Note that for the tropics the detection of fronts is quite questionable. However, for comparison with the TFP method and evaluation of Catto and Pfahl (2013), these regions are included, although the front detection methods work only in the extratropics in a meaningful way. Nevertheless, our results are in good agreement with those derived in Catto and Pfahl (2013).

**Extreme precipitation associated with fronts relative to frontal frequency:**

In Fig. 11 we display $R_1$ as a function with respect to $P_{a(fr)}$. We divided all sample points into $k = 21$ bins. Each bin $b_i$ with $0 \leq i < k, i \in \mathbb{N}$ contains all $R_1(p)$ for all grid points $p$ within the midlatitudes, excluding high altitudes, where $(i-1) \cdot 5\% < P_{a(fr)}(p) \leq (i) \cdot 5\%$. Additionally we plotted the fitted $1st$ and $99th$ percentile as well as the Identity as orientation. The lines and boxlots can be interpreted as follows: If the box plot is above the $99th$ percentile line, we can assume that the correlation between extreme precipitation events and fronts is significant in terms of our test, described above.

For warm fronts, cold fronts and occlusions we can see that both the median and the mean of each bin very quickly grow larger than the $99th$ percentile, which indicates a significant connection between fronts and extreme precipitation. For stationary fronts this appears less clear. Up to 20% the curve connecting the medians appears to show a significant correlation between extreme precipitation and stationary fronts, before suddenly flattening. Regarding the plot showing the results for all types of fronts, we can see that for all bins, except the last, the mean and median clearly exceed the $99th$ percentile. This clearly indicates a strong connection between fronts and extreme precipitation.

**Fronts associated with extreme precipitation:**

In the previous part we have shown that a high percentage of extreme precipitation can be associated with a front. We also saw that outside the tropics this connection is significant, with respect to our statistical test. However for a clear image we are also interested in the proportion of fronts that can be associated with an extreme Event ($R_2$). Similar to Fig. 10 we

**Table 9.** Average Proportion of extreme precipitation events associated with a front for different regions in 2016. global $[-60°60°]$N, northern and southern hemisphere $[0°, 60°]$N and S respectively, tropics $[-30°, 30°]$N.

| Region | all | warm | cold | occlusion | stationary |
|---|---|---|---|---|---|
| global | 0.591762 | 0.207308 | 0.259069 | 0.137746 | 0.152227 |
| northern hemisphere | 0.523959 | 0.158889 | 0.205674 | 0.115030 | 0.176706 |
| southern hemisphere | 0.658888 | 0.255434 | 0.312175 | 0.160145 | 0.127472 |
| tropics | 0.419067 | 0.074942 | 0.144921 | 0.023774 | 0.225288 |
| global land | 0.426572 | 0.097443 | 0.168555 | 0.080147 | 0.186018 |
| global sea | 0.665551 | 0.256384 | 0.299502 | 0.163476 | 0.137133 |

plotted $R_2$ per grid point in Fig. 12. Light gray and white regions are masked as before, while this time regions where no front of the corresponding type occurred are masked dark gray. The first striking observation is that regions where a front is less likely to occur tend to have a higher percentage of fronts being associated with extreme precipitation. This very clearly shown for the occlusions, where occlusions occurring close to $30°N/S$ are almost always associated with extreme precipitation. In general for the midlatitudes up to more than $40\%$ of fronts can be associated with extreme precipitation. The decrease in $R_2$ for regions with a higher $P_{fr}$ can at least partially be explained by the definition of extreme precipitation, as it inherently limits the amount of extreme precipitation events. If $P_{fr}$ exceeds that amount it is likely that several fronts may not be associated with an extreme precipitation event, even though strong precipitation still occurs. This is somewhat dampened by the fact that $R_2$ uses $P_{a(epr)}$ giving each front several grid points to be associated to. Compared to Catto and Pfahl (2013) our results show the same tendencies. Nonetheless our results indicate that a higher amount of detected fronts can be associated with extreme precipitation.

Overall our results show a significant connection between extreme precipitation and detected network fronts. Our results generally agree with the results of the previous study by Catto and Pfahl (2013). This once again highlights our networks potential to be used in future scientific research. We additionally investigated the correlation between fronts and extreme precipitation on a higher resolution, i.e. for two smaller radii of $5px$ $(1.25°)$ and $2px$ $(0.5°)$, respectively. The qualitative features (i.e. the regions with high probability) remain the same but the frequency of occurrence is reduced due to the smaller radius of influence. The respective figures can be found in the SI as Fig. S4. Such investigations cannot be carried out with classical TFP methods, since they are restricted to low resolution data sets. This underlines the benefit of our new method over existing ones.

## 4 Conclusions

Atmospheric fronts are important features, which are usually associated with synoptic scale weather systems. Since fronts are usually connected with significant weather, i.e. clouds and precipitation, and occasionally with extreme precipitation events,

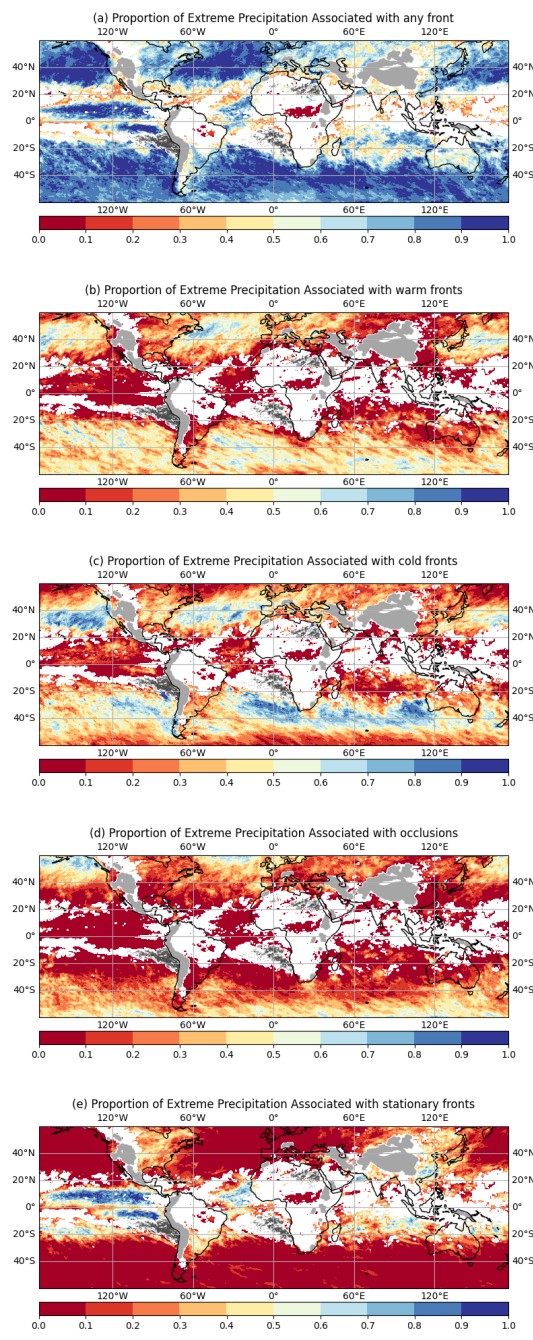

**Figure 10.** Proportion of extreme precipitation events, which are also associated with a front. High orography is masked as light gray, while areas where no extreme precipitation events occurred in 2016 are masked as dark gray. Regions where no significant correlation between extreme precipitation and fronts was found are blanked. (a) any front, (b) warm front, (c) cold front, (d) occlusion, (e) stationary front

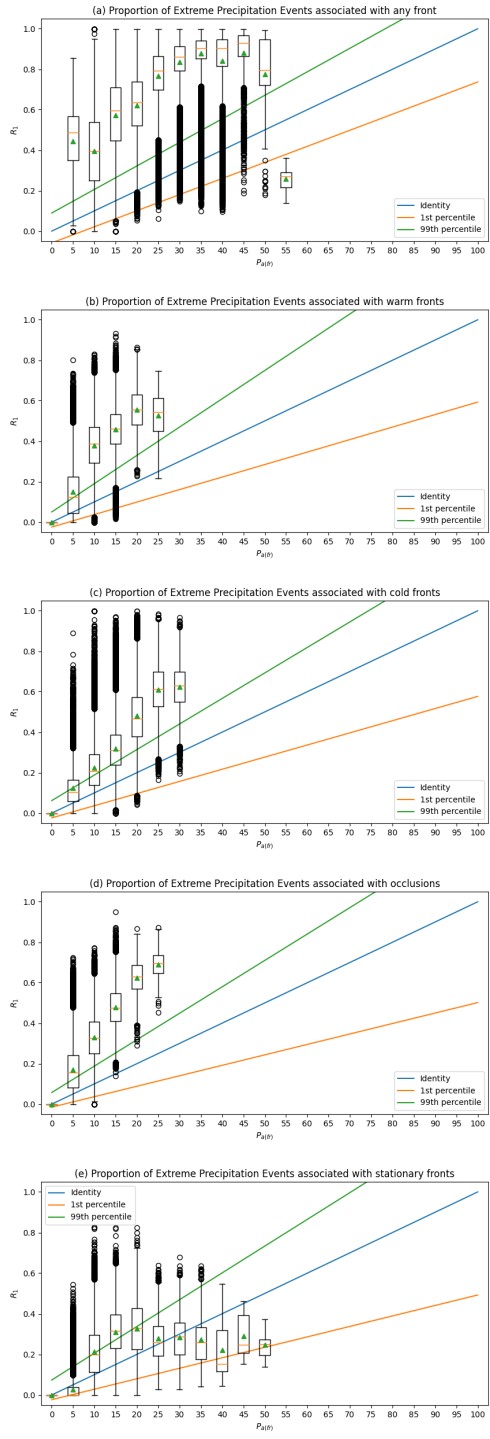

**Figure 11.** Fraction of extreme precipitation events grouped by frontal frequency as boxplots. Including $1st$ and $99th$ percentile of the statistical test. (a) any front, (b) warm front, (c) cold front, (d) occlusion, (e) stationary front

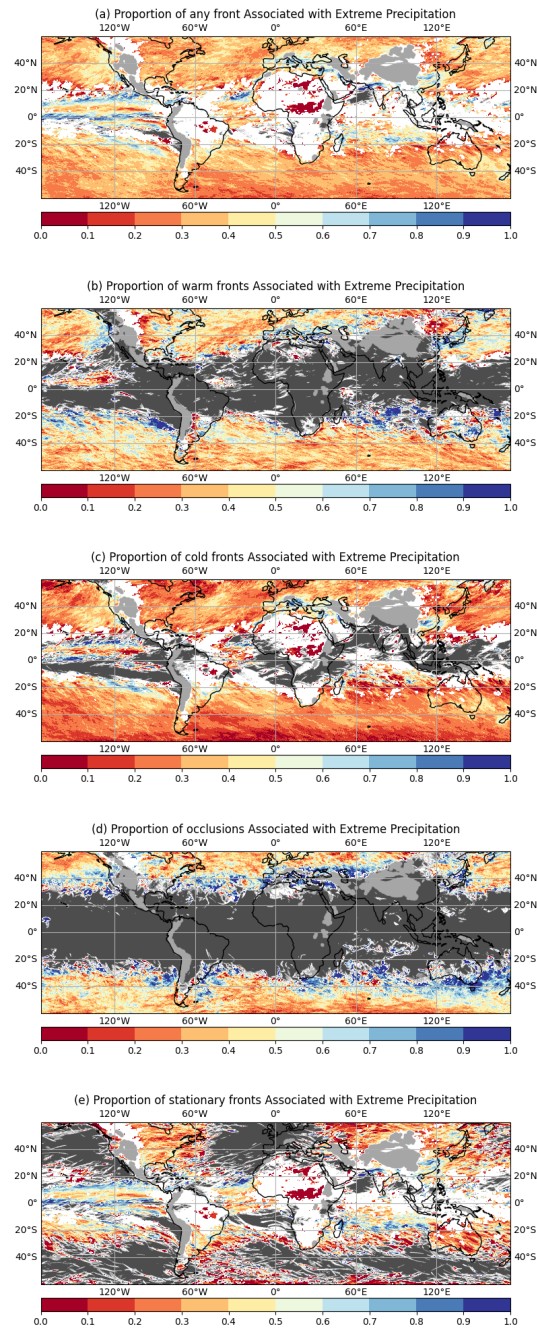

**Figure 12.** Proportion of fronts, which are associated with an extreme precipitation event. High orography is masked as light gray, while areas where no fronts of the corresponding class where detected in 2016 are masked as dark gray. Regions where no significant correlation between extreme precipitation and fronts was found are blanked. (a) any front, (b) warm front, (c) cold front, (d) occlusion, (e) stationary front

**Table 10.** Average Proportion of extreme precipitation events associated with a front for different regions in 2016 for the midlatitudes $[30°, 60°]$N and S respectively.

| Region | all | warm | cold | occlusion | stationary |
|---|---|---|---|---|---|
| midlatitudes | 0.761661 | 0.337388 | 0.372848 | 0.248444 | 0.080310 |
| northern midlatitudes | 0.678892 | 0.270840 | 0.311021 | 0.212936 | 0.133108 |
| southern midlatitudes | 0.843307 | 0.402863 | 0.432948 | 0.284470 | 0.027839 |
| midlatitudes no mountain | 0.780816 | 0.354091 | 0.383997 | 0.260504 | 0.071064 |
| midlatitudes sea | 0.851108 | 0.415874 | 0.425085 | 0.295962 | 0.029676 |
| midlatitudes land | 0.565787 | 0.165520 | 0.258460 | 0.144388 | 0.191187 |
| midlatitudes land, no mountain | 0.596549 | 0.192130 | 0.276355 | 0.167556 | 0.179444 |

they are of high interest for weather forecasts but also in terms of scientific research of such events. While the term front refers to a sharp transition between air masses of different characteristics (e.g. in terms of temperature, humidity etc.), there is unfortunately not a generally accepted definition of a front. This is also reflected in many different approaches to detect fronts automatically, e.g. using (multiple) gradients of thermodynamic variables, or even recently using machine learning techniques. In this study we present a new method for automatic front detection based on a neural network, which uses ERA5 reanalysis data. As a ground truth for training and validation, we use surface front data from two different weather services (NWS and DWD), covering significant parts of the Northern hemisphere; for validation a disjoint subset of this data set is used. We train the network on a loss function, that allows to classify and predict fronts across the input regions. Our applied loss function leads the network to predict clearly localized fronts without the need of morphological post processing thinning operations. The network is able to predict fronts with a Critical Success Rate higher than about 66.9%, and an Object Detection Rate higher than about 77%. For a better evaluation of the quality of the method, we compare the network output with a baseline method, which uses a traditional approach of thermodynamic variables (TFP approach). For both methods a climatology of fronts is derived. In this direct comparison, the new method outperforms the baseline method in the direct comparison with the data from the weather services. We can show that we cannot simply transfer a locally trained network onto any other region but rather need to train on several data-set to obtain a reliable general front detection. The climatology results indicate that a transfer on oceanic regions maybe feasible, however this has to be evaluated in future research. It is also desirable to further investigate up to which degree extrapolation onto different regions is possible and to investigate whether or not generalization onto global data is possible from just a few subregions. Our evaluation of physical properties accompanying our detections show that our detected fronts generally exhibit similar properties as those usually looked for in classical methods. As an example gradients in the equivalent potential temperature are shown. In addition, a similar quantity as for classical TFP methods is determined from equivalent potential temperature. In the comparison for these quantities at fronts determined by the weather services and detected by the network, respectively, we find very good agreement; in addition, these fronts exhibit the same features as would be detected by a TFP method. This also shows that our ground truth data, surface fronts originating from two weather services,

is a suitable choice; although surface fronts are detected, they show the correct structure in terms of thermodynamic variables. Thus, surface fronts can serve as a proxy for the detection of fronts, however our analysis shows that the resulting fronts are meaningful. In a final application, we investigate the connection of fronts with extreme precipitation events. This investigation

is guided by the former investigation by Catto and Pfahl (2013); however, our network allows us to fully use the available resolution of ERA5 and possible research characteristics of fronts at a high spatial and temporal resolution, leading to a more detailed investigation. The correlation of extreme precipitation events and fronts can be determined. For the midlatitudes the effect is most prominent, the strongest correlation can be seen for fronts and events over flat terrain, especially over the ocean. This application shows that this new method is not only just a tool for operational weather forecasting, but also a serious

method for scientific investigations. Since the method can be applied on high resolution data, this shows the benefit of the new method over the existing TFP methods, which are usually restricted to low resolution data set. The method is quite flexible, it is quite straightforward to include new training data sets, as e.g. surfaces fronts for the southern hemisphere. In addition, there is no principle obstacle for using meteorological data sets with higher resolution as input for the method. In future work separating the detection from the classification task may be beneficial, seeing the good detection rates of the presented network

in the binary case. We would also like to further explore the application and effect of other methods to handle the label bias, such as the method described by Acuna et al. (2019). In terms of research in the field of meteorology, we want to apply this method for further research on the connection of frontal systems with other phenomena, e.g. for the investigation of clouds at different heights around fronts or transport phenomena associated with frontal systems.

*Code and data availability.* The latest code is available at https://github.com/stnie/FrontDetection. A doi will be submitted later. ERA5

Reanalysis data can be accessed via the ECMWF climate data center. Used NWS frontal label is available with doi: 10.5281/zenodo.2642801 (National Weather Service, 2019). Access to the DWD data may be granted by the DWD.

*Video supplement.* A video supplement showing predicted fronts for January 2016 is available at https://av.tib.eu/media/54716 (Niebler, 2021)

*Author contributions.* Stefan Niebler implemented, and trained the network. He also evaluated the baseline method as well as the network.

Bertil Schmidt, Annette Miltenberger, Peter Spichtinger, and Stefan Niebler wrote the draft of the manuscript. Bertil Schmidt, Annette Miltenberger, and Peter Spichtinger proposed and supervised the project. All authors edited the manuscript and analyzed the results.

*Competing interests.* The authors declare that they have no conflict of interest

*Acknowledgements.* The study is supported by the project "Big Data in Atmospheric Physics (BINARY)", funded by the Carl Zeiss Foundation (grant P2018-02-003). We acknowledge the ECMWF for providing access to the ERA5 Reanalysis data. We further acknowledge the ETH Zurich and especially Michael Sprenger for providing the code for the used baseline method. Label data for the European continent and Northern Atlantic was provided by the Deutscher Wetterdienst. Label data for the North American continent was provided and made publicly available by the North American Weather Service. We further acknowledge the ZDV of the Johannes Gutenberg University and the Mogon II Super Cluster for providing the necessary hardware and computing time to execute our experiments. We thank Philipp Reutter and Holger Tost for fruitful discussions.

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
