# Peer review of "Automated detection and classification of synoptic scale fronts from atmospheric data grids"

_Weather and Climate Dynamics, 2021_

## Author Response (AR1)

**Automated detection and classification of synoptic scale fronts from atmospheric data grids**
**Response to Referees and Co-Editor**

October 15, 2021

**1 General response**

First of all, we thank both reviewers for their helpful comments and suggestions which lead to a significant improved manuscript. This document contains our responses to all reviewer comments.

**2 General Changes**

- We updated the manuscript in several parts.We moved the parts about trained models, CSI calculation and how the evaluated output is created from the old Section Results into Section 2.4.

  We integrated the Discussion directly into the new Section 3: "Results and Discussion" and therefore removed the Section Discussion.

  Section 3 now consists of two parts. The first section 3.1 is related to a quantification (CSI (section 3.1.1) and Correlation of Climatologies (section 3.1.2)) of our networks results compared against the weather service labels. This includes a comparison against the baseline method.

  The second part (section 3.2) is a more qualitative evaluation, where we inspect the average change of physical quantities across our detected fronts at 850hPa and evaluate those observations against theoretical expectations such as the front criterion of TFP methods (section 3.2.1).

  Additionally we evaluate a study linking our detected fronts with extreme precipitation events similar to Catto and Pfahl, in return showing how our method can be applied in a scientific context (section 3.2.2).

- We updated the Evaluations: We refined the CSI evaluation, to reduce the effect caused by the input cropping. A new description of the CSI calculation is found in Section 2.4.3 For the Climatology we restricted our quantitative evaluation to the midlatitudes. (Table 7 and 8, Figure 7)

- We evaluated the Frontal Cross Sections at 850hPa and also evaluated different variables, including the relative humidity and the equivalent potential temperature and its gradients, for a better intuition of the quality of the detections, with regard to classical methods such as TFP based methods. (Section 3.2.1).

- Further we created a new Video Supplement, showing detected fronts against the equivalent potential temperature instead of specific humidity.

- We put more emphasis on the used Loss function and Label deformation and updated these sections to make them easier to understand. (see Sections 2.2.4 and 2.2.5)

**3 Referee 1**

This is a good, well-written paper that should be of interest to the readership. However, I have a couple of minor comments that could be addressed in a revised version of the paper.

**Comments:**

1. The paper is longer than it needs to be, and some information is spread over the paper which makes it difficult to extract the relevant pieces. E.g. when introducing the vertical levels in l.117 and then

mentioning that you only use 9 pressure levels in l.196. Why not describe the data-set augmentation with the data in section 2.1?

We have rewritten some parts of the paper and tried to condense some redundant parts of the manuscript. We reordered some sections, put the discussion of the results into section 3, deleted the discussion section, and added some more text to the conclusion.

2. Section 3.1 and 3.2: I have tried for a while to understand why you present results for the validation AND the test dataset and gave up. Why do you need section 3.1? You are writing in l.300: "We validated our model during training using 1460 samples of data from 2017. We evaluated our trained models on 1 year of data from 2016 using an object based evaluation described as described later in this section." This does not really explain why you need the two sections. Also, section 3.1 starts with "The trained models were evaluated on test sets..." which generates ultimate confusion. Do you loose any information when removing section 3.1? Maybe I am missing something.

We agree that the information from Section 3.1 does not add any important information to the overall manuscript. In combination with the length of the Manuscript and the fact that we plan to add an application of the method, we decide to remove Section 3.1. Additionally we will rearrange some information regarding the evaluation from this section into the methodology section to remove the spreading.

3. l.193: I do not understand this. You say you "ignore the outer 20 pixel". But then you are saying that the brighter areas can be used as input in the caption of Figure 4. Are they used as input but not predicted? But then the output domain should be smaller than the input domain in Figure 3...? And why do you crop to 128x256 pixel (l.199)? And then there is again a confusing mentioning of the 5 degree border in the caption of Table 2...

Yes, the bright+dark shade describe possible input regions, the dark shades are the regions, where a valid output can be located. The network essentially provides an output of the same size as the input, however as outer pixel may have far less information, we decide to ignore the outer 20 pixel of the output, such effectively reducing the size of the output region. So yes for a given input domain, the output domain is smaller, due to this. The $128 \times 256$ crop is performed to ensure that our networks input is divisible by 8 as well as to create some more samples to draw from during training. We rewrote the respective section to make it more clear, why and how this is done.

4. l.8: I would not call the baseline model "ETH". ETH is a very large institution.

We now refer to the method as "baseline"

5. l.21: Maybe add a reference to the Mei-Yu front?

We added a suitable reference

6. l.22: "Determining the position and propagation of surface fronts plays an important role for weather forecasting". Well, the prediction of the position, yes. But is the same true for the automatic detection? Fronts can easily be identified in field maps by the trained eye. Why do we need the ability to detect them automatically with ML? I do understand why, but it would be good if this would be made more explicit in the intro, otherwise it seems that you have a hammer and are searching for nails.

The detection of fronts in operational weather forecasts is of course an important task. However, even for postprocessing of model data this would be interesting. Beside the task of operational weather centers, an automatic classification of fronts in meteorological data set is of interest for research purposes. As we show in the newly added application, we can use the automatic detection for statistical evaluations, e.g. for the connection of fronts with extreme precipitation events. Other examples are obvious, as e.g. the connection of clouds and convection to fronts of different types. We added some text in the manuscript to highlight this purpose.

7. l.24: What are empirical guidelines?

Many weather services as the DWD have some guidelines how to determine fronts. Of course, the physical variables play a role. However, also some empirical connections and features might be helpful for certain regions to determine fronts and other features manually. This was personal communication with the DWD. Unfortunately, we cannot provide detailed examples. We deleted the word empirical, since it is misleading.

8. Section 2.1: Maybe I missed it, but do you actually state the resolution of the NWS and DWD datasets somewhere (or the resolution equivalent of the PNG image)?

NWS labels are given as coordinate pairs with a 0.1° resolution, DWD images come at a $4389 \times 3114$ pixel resolution, from which we extract coordinate pairs. We added this information to the paper

9. Figure 3: I do not understand the encode and decode blocks. Can you add some info here? Also, what are the white boxes the "copy" arrows end in?

   The code and decode blocks are just several convolution, ReLU and BatchNorm operators in sequence. We decided to put references to those at the corner of the image, to not unnecessarily convolute the network architecture image. The white boxes are the results of the copy operators. Basically: The box at the beginning of the gray arrow is copied along the arrow and the destination of this copy is described by the white box. This white box is concatenated to the result of the upsample operation coming from below. We tried to make this more clear in the manuscript, by adding some text, also in the figure caption

10. l.198: "If both labels are available". What does this mean? At a certain point in time? Why should this matter?

    Yes, we refer to a certain point in time. As our data sets from the NWS and DWD do only overlap for a subset of our data set, we will have input data, where only a label from one weather services is available. In the case that we have a label for both weather services for a certain point in time, we randomly select which one to use. The result of this selection can vary between epochs. On the other hand if only a single label is available, we do not need to randomly select, which label to use, as there only is one. We updated the respective section in the manuscript

11. Table 2: The whole caption should be reformulated. "For the global region this border is included within the mentioned range." ?

    We made a new table, where we explicitly state the input and output regions used during training.

12. l.242: This paragraph is important but very difficult to understand. It should be rewritten.

    We have rewritten the paragraph and added a figure to visualize this approach. We also added more details about the method.

13. l.279: I would not use "t" for the index of the channels as "t" is often used for time.

    We have changed the index

14. l.280-282: I do not understand this. "individually for each batch"? "more emphasize onto classification"? Either equation (2) holds, or not.

    The loss weighs how many samples in a batch contain certain labels, which is why the weights may differ between batches. We have rewritten this part and updated the loss to clearer represent that the weights used for the loss are dependent on the available labels in the batch.

15. l.289: Why did you not evaluate the baseline at 0.25 degree? I guess there are good reasons, but please state them.

    The baseline natively runs at 1 degree. An upscaling of the method to 0.25 degree was unfeasible, because of the additional small scale features, disturbing the gradients of the thermodynamic variables. Simply upscaling the results would introduce additional ambiguities in the placement of the labels. A downscaling seemed to be the more accurate choice at this point. We added some text about this into the manuscript.

16. Table 3: You can as well remove the "Stationary" line.

    We removed it.

17. Table 5: "The suffix "all"..." I do not understand this sentence.

    We show two slightly different evaluation methods in the manuscript, which were also applied to this evaluation. In the updated version we only use one of these when comparing against the baseline method, which we explain in the previous section. This ultimately means that we do not need the distinction in this table. We therefore removed the sentence.

18. l.488: I find this a bit confusing. You would not leave out a certain region in a real-world application, so why here?

    As Referee 2 noted, the baseline method is not suited for application outside the midlatitudes. As Greenland is located outside the midlatitudes, it shouldn't have been in the quantitative evaluation for a fair comparison. In our previous evaluation it was apparent that Greenland caused a lot of false positive

fronts detected by the baseline method which drastically reduces the correlation coefficient. Removing Greenland from the evaluation was to show that the low score was mainly caused by this comparatively small region, while the rest is pretty accurate. Nonetheless, as stated we decided to restrict this evaluation to the midlatitudes, which in return already excludes Greenland making this distinction obsolete.

Typos etc:

1. l.51: typically → done

2. l.253: predicted fronts → rewritten

3. l.301: remove "described" → done

4. l.346: "be be" → removed

5. l.349: "slight edge"? → rewritten

6. l.351: "fact that training" → done

7. l.403: "most likely" → done

8. l.445: "and the European data" → wrote "than the European data"

9. Caption Figure 7: "on the for the"→ done

10. l.514: "for is the lack" → added a this

11. l.439: "However," → done

**4 Referee 2**

Automated feature recognition has proven useful in gaining scientific knowledge of the dynamics and relationships between various atmospheric flow features such as cyclones, jets, and surface fronts. However, there are a variety of automated methods to identify the relevant feature of interest because even trained experts do not agree on how to define a feature. This applies to surface fronts for which no general definition exists. Therefore, improved methods that help to gain additional insight into the nature of fronts are important, however I am have some concerns on whether ML-based method trained on surface analysis is the next step.

**General comment**

1. There is no single accepted front definition and different weather centers use their own definitions based partly on physical considerations, partly on training and experience, partly on the specific local meteorology, and sometimes simply artistic. It is therefore questionable whether a front identification should be guided by manual surface maps or physical arguments. This dilemma is nicely summarized in Uccellini et al (1992) and Sanders (1999) and was lately reviewed in Schemm et al. (2018) and Thomas and Schultz (2019a). I recommend that the authors review these earlier studies; their introduction comes in its first paragraph without a single reference (and there are numerous studies that link fronts to extreme weather that could be referenced). Also, there is little historic background provided.

   The front dilemma can be summarized with the following example: The UK MetOffice automated surface analysis regularly displays double fronts, while the DWD chart never shows these fronts – see Fig. 2 in Thomas and Schultz (2019a). Instead, DWD-front are Norwegian-like and hemispheric spanning, which is more art than science. The missing double fronts are however real and important to detect. They will be missed if trained on DWD charts.

   Related to the definition of fronts, there is one stream of front definitions that is based on baroclinic instability and there is also a second front definition based on air-mass boundaries – see Thomas and Schultz (2019b). The two are mixed up in this study, for example, when the authors speak about fronts that are associated with the propagation of extratropical cyclones but thereafter describe fronts as air-mass boundaries. These air-mass boundaries, which provide little baroclinicity, are very interesting for research. If one wants to detect these, one cannot train a ML method on DWD charts, although it seems as if DWD uses many meteorological variables to draw their front, which is common for the air-mass boundary definition of fronts but not that based on baroclinicity. Maybe DWD excludes mesoscale fronts in general?

Overall, I therefore reject manual surface charts as ground truth, baseline method or "gold standard" for verification. The surface maps are biased, inhomogeneous, only partly based on physical reasoning and cannot be transferred between different regions. I find a tool that learns these biases, here the DWD bias, difficult to use for research purposes, though they might be useful in an automated DWD workplace environment. Even though the authors try to alleviate some of these issues with the blurring of the front position shown in their Fig. 5, I hesitate to conlcude that ML-based fronts trained on surface charts is the way forward.

We respectfully disagree that manual surface charts are not suited as a baseline. In our (updated) evaluation we show cross sections of both our detected fronts and the provided weather service fronts at 850hPa, especially for equivalent potential temperature, as used e.g. for the baseline method (Figure 8 and 9, section 3.2.1). Both fronts seem to show the characteristics that more traditional TFP methods are based on, e.g. TFP = 0. This alone seems to refute the believe that the surface fronts do not follow any physical reasoning. Further you state that the weather service label contain personal bias of the executing meteorologists and acknowledge that we try to alleviate some of these biases. However you misunderstood that we are trying to inflate the label. We do use a label deformation procedure adjusting the labels prior to evaluation (see Section 2.2.4). The idea behind this was that it would help to relocate biased labels, assuming that they are at least somewhat correctly placed. As mentioned before the cross sections seem to indicate that this is indeed the case, which is why we believe that those are suited training data and the output of the trained network is a suitably good front detection. At last we also use **two** different weather services (NWS and DWD) for training, which should reduce the individual bias of each of those sets by simply introducing more meteorologists in our training data. Actually, we can show that the performance of the method increases if we use training data from more than one weather service. Finally, we want to note here that even if the training data stemming from DWD does not include double fronts, a closer inspection of the movie in the SI shows that indeed the network can provide double fronts, e.g. fronts in the North Atlantic at 0:39 and 0:45, as well as in the South Atlantic at 0:30 east of North and South America. Obviously, the network is able to generalize the features even if these fronts are not explicitly included in one training data set.

2. The manuscript has a strong technical nature with only little insight into meteorology or front dynamics. I would recommend considering a transfer of this manuscript to GMD.

While there are other papers within WCD that are of similar technical nature, we did add an application of our trained network researching the connection of extreme precipitation and our detected fronts using the ERA5 data and 1 hourly aggregated precipitation, similar to a study by Catto and Pfahl.(Section 3.2.2) Therefore, we think that WCD is still an appropriate choice for the manuscript.

3. The presented comparison against a second front detection method, which is based on the thermal front parameter (TFP), is odd. First, I recommend not to call it "the ETH method", because this is not known to the community and ETH is a large institution. Maybe TFP method? Second, I recommend providing more background about the TFP, which goes back to Renard and Clarke (1965). The TFP implementation by Jenkner et al (2010), which is used here as reference method, is unique, because it places the front where TFP=0, which is at the center of the frontal zone. However, this is not where most meteorologist place the front. Most center, including DWD and ECMWF, place it where MAX(TFP)=0, which is at the leading edge of the frontal zone. So, there is a mismatch. This important difference is not explained in the current manuscript and because the width of a frontal zone can easily encompass a couple of hundred kilometers, the here used "baseline method" will be due to its design in most situations do not agree with the DWD charts. Basically, a method that was trained to reproduce DWD fronts, which it does very well, is compared against a method that was intentionally designed not to agree with DWD fronts because the front line is placed in a different location. It is thus not a meaningful comparison (and this explains much of what is found in Lines 376-4079) and I recommend that the comparison is removed.

We changed the name of the method into "baseline method". We added some information that the placement of the baseline TFP method may be offset compared to the weather services, to make it clear to the reader (e.g. Line 618 in section 3.2.1). However, in our CSI evaluation we do not calculate the exact matching of fronts but rather use a more soft criterion. To match fronts we use a search radius of $D = 250\,\text{km}$. To be considered a match only the median distance of a detected front from the closest pixel of a provided labeled front hast to be below $D$. We thought that this is an already loose enough criterion to match fronts, however we additionally added another evaluation where we evaluate the baseline TFP method using $D = 500\,\text{km}$ essentially doubling the search radius by adding another $250\,\text{km}$ to the radius (Table 6). Also we would like to remind you, that we did not solely use the DWD data for evaluation

but also that of the NWS, i.e. from two weather services. Actually, the method is very flexible, such that additional training data sets can easily implemented.

4. More meaningful would be a comparison against another ML-based method, for example, that of Lagerquist et al. (2019), who pioneered ML-based front detection. This would be more insightful because it is not clear at this point which neural-network architecture is most suitable for front detection and why this is the case. I find Fig. 10 in Lagerquist et al. (2019) very helpful. A similar figure plus a direct comparison of these two ML-based methods would thus be of high merit.

We tried using the method, however we could not get it to run in a feasible time. Additionally the method is executed and evaluated on a different grid and spacing which makes a direct comparison inaccurate at least.

5. It is not advisable to transfer an automated front detection from a low-resolution grid to a high-resolution grid without intensive retuning and testing. How was this retuning done? By how much was the detection thresholds for the front gradient increased? By how much was the minimum length criterion changed? Did the authors increase the minimum advection speed to separate stationary from non-stationary fronts? A method developed for a ERA-Interim 1x1 degree grid (or for a 2-km grid as in Jenkner et al. 2010) should not be transferred to another grid spacing. Further, does DWD use a minimum front length and are the authors using the same threshold? At the same time, while the front threshold presumably was increased when going from a 1x1 degree grid to a 0.5x0.5-degree grid, the number of fronts ideally should not change (see Fig. 2 in Thomas and Schultz (2019b) on the dependency to the threshold). More details on the retuning that was done when preparing the comparison is needed in this manuscript.

For the tuning of the algorithm we tested different settings for the number of parameters in the algorithm, namely the minimum advection speed ($3\,\mathrm{m\,s^{-1}}$ to $6\,\mathrm{m\,s^{-1}}$), the minimum temperature gradient ($4 \times 10^{-2}\,\mathrm{K\,km^{-1}}$ to $5 \times 10^{-2}\,\mathrm{K\,km^{-1}}$), the minimum front length ($500\,\mathrm{km}$ to $700\,\mathrm{km}$), the number of gridpoints a front object has to contain (15 to 50), the value of a smoothing parameter for frontal lines (5 to 30), the gap allowed between two segments of large THE gradient from them to be considered as one front object ($5\,\mathrm{km}$ to $100\,\mathrm{km}$), and the number of times a digital filter is applied to the equivalent potential temperature gradient field (5 to 10). For the tuning exercise we considered the three month of December 2013, January 2014, and February 2014, for which both the ERA-Interim and the ERA5 reanalysis are available. For the three month in total 1947 different climatologies were computed and compared by considering the number of fronts detected in the extratropics ($-60°$ to $-30°\,\mathrm{N}$ and $30°$ to $-60°\,\mathrm{N}$), the geographical location, the length of fronts detected, and a visual inspection of individual cases from similar performing parameter combinations.

In the original version of the algorithm those parameters, i.e. that applied to the ERA-Interim data at $1°$ grid spacing, are set to $3\,\mathrm{m\,s^{-1}}$, $4 \times 10^{-2}\,\mathrm{K\,km^{-1}}$, $500\,\mathrm{km}$, 15, 5, $100\,\mathrm{km}$, and 5. We decided to keep as much of the physical values, which are determined in units of $\mathrm{km^{-1}}$ or km identical to the original algorithm. The motivation is to retain similar physical properties of the front. Except from smoothing of gradients on a lower resolution grid, it is expected that the definition of the parameters in units of km or $\mathrm{km^{-1}}$ already reflects the change in grid resolution. However, we adjusted the number of the minimum number of grid-points in a front object (changed from 15 to 20), the number of filter applications (changed from 5 to 7), and the smoothing parameter (changed from 5 to 15) to reflect the larger spatial resolution of the ERA5 data-set (here: $0.5°$). These parameters determine the smoothing of the equivalent potential temperature gradient field and the "straigthening" of frontal lines and hence we deem them more appropriate for adjustment to different grid spacings.

Fig. 1 shows the number of fronts detected in the northern and southern extra-tropics for different experiments. In all the shown experiments the filter is applied 7 times to the equivalent potential temperature gradient field, as fewer applications lead to large increase in the number of detected fronts and a shift towards short and disconnected frontal features and a more applications to a large decrease in the number of detected fronts. The number of fronts in the ERA5 data-set with the re-tuned algorithm the number of detected fronts is about $30\,\%$ larger than in the original data-set, which could be remedied by increasing the minimum length of fronts to $700\,\mathrm{km}$ or increasing the minimum potential temperature gradient to a value between $4 \times 10^{-2}\,\mathrm{K\,km^{-1}}$ and $5 \times 10^{-2}\,\mathrm{K\,km^{-1}}$. Considering also the distribution of front length (not shown) and the spatial distribution of front occurrence (see Fig. 2), we decided to not change the values of minimum front length and the minimum potential temperature gradient as both do not yield a benefit in terms of the spatial front distribution.

The information, which parameters of the front detection algorithm have been adjusted has been included in the revised version of the manuscript. (Section 2.3)

6. I was disappointed to see an equivalent-potential temperature based front definition purposefully applied

[Figure]

Figure 1: Number of front objects identified in the northern and southern extra-tropics from the ERA5 data-set for different choices of the free parameters. Green symbols correspond to climatologies computed with a minimum THE gradient of $4 \times 10^{-2} \, \text{K} \, \text{km}^{-1}$ and cyan symbols to those with a minimum THE gradient of $5 \times 10^{-2} \, \text{K} \, \text{km}^{-1}$. Within each group experiments are grouped according to minimum advection velocities of $3 \, \text{m} \, \text{s}^{-1}$, $4 \, \text{m} \, \text{s}^{-1}$ and $5 \, \text{m} \, \text{s}^{-1}$ (groups of points separated by dashed lines) and minimum front length of $500 \, \text{km}$, $600 \, \text{km}$ and $700 \, \text{km}$ (groups of points between the dashed lines). The blue horizontal line shows the number of fronts detected with the original algorithm in the ERA-Interim data-set.

to latitudes outside of midlatitudes. Front detection methods based on equivalent-potential temperature (called TH in the next statement) are well known to be unsafe for usage outside of the midlatitudes, for example, Schemm et al. (2015, p. 1696) noted: "... clearly indicates that the TH method is influenced by semi-permanent convergence zones and tropical convection (although a minimum advection threshold is applied). Tropical features which, from a synoptic viewpoint, would not be regarded as a 'front', are identified as such. Accordingly, TH methods should be used with care if applied outside midlatitudes", which is a nice way to say that it should not be done. Further they note "... as the $\theta e$ gradient can be dominated solely by moisture gradients, especially in tropical latitudes, this results in the detection of several quasi-stationary fronts (which form along mountain crests, or in association with land – sea contrasts) which must be removed in a post-processing step" (Schemm et al. 2015, p.1687). Against these recommendations the authors decided to apply the $\theta e$ method to subtropical and tropical latitudes and afterward, not too surprisingly, conclude that it detects numerous of non-cyclone related fronts. What was the intention behind this? The section between L.460-470 is therefore misleading.

We understand that the used baseline method is not suited for the application outside the midlatitudes. As a result we will restrict our quantitative evaluation to the midlatitudes, as a fair way to evaluate the performance. But we still believe that an a qualitative evaluation outside the midlatitudes is of interest to highlight the differences in how the network performs in the regions where the baseline method struggles / should not be used. We do agree with you that we should make it more clear that the baseline is not designed for these regions. To do this we added some text when describing the baseline as well as adding a gray overlay to the climatology images to indicate this.(Section 3.1.2, Figure 7). Finally, we want to add that although we are aware of the restriction of front detection methods for the tropics, we added the results for the tropics about the connection between extreme precipitation and fronts. This inclusion was also motivated by the results of Catto and Pfahl (2013), who also applied a TFP mehod in the tropics for their comparisons.

7. The conclusion is short, with only a technical statement and an outlook but no conclusion related to weather and climate dynamics. Maybe you could try to conclude on how and why the ML-based method is able to distinguish mobile from stationary fronts (such as those along the coastlines or mountains), which would yield additional process understanding and it is a mayor struggle for traditional TFP-based methods.

We reordered some sections, put the discussion of the results into section 3, deleted the discussion section, and added some more text to the conclusion. We added another chapter regarding the connection between our detected fronts and extreme precipitation (section 3.2.2). A direct explanation of such a deep learning architecture is hard and it is a very current field of research, to get reliable conclusive information from a neural network. Showing the physical cross-sections indicates that the algorithm respects the wind speed close to stationary fronts to detect those, seeing how it is far lower than the other types of fronts (Figure

[Figure]

Figure 2: Number of times a front is detected at a specific grid-point in the ERA-Interim data-set (top row) and the ERA5 data-set using different minimum THE gradients (middle row: $4 \times 10^{-2}\,\mathrm{K\,km^{-1}}$, bottom row: $5 \times 10^{-2}\,\mathrm{K\,km^{-1}}$) and different minimum frontal length (left: $500\,\mathrm{km}$, right: $700\,\mathrm{km}$).

9). However it is hard to detect whether this is causation or simply correlation.

Minor comments:

1. L. 19 "much of the literature is on the larger-scale fronts" – research on mesoscale fronts is a very active field of research as well.

   We added some text into this direction

2. L. 15 "are a vital part of the communication of weather to the public and the public perception of weather in general" – Most people use Apps; fronts are no longer a major part of modern weather communication.

   We added a comment on this

3. L. 27: "The former methodology goes back to the work by Hewson (1998)" – I guess it goes back to Renard and Clarke (1965).

   We added the reference and some text

References:

• Lagerquist, Ryan, Allen, John T., and McGovern, Amy, 2020, "Climatology and Variability of Warm and Cold Fronts over North America from 1979 to 2018" Journal of Climate Vol. 33, No. 15, 1520-0442

• Lagerquist, R., A. McGovern, and D. Gagne II, 2019: Deep learning for spatially explicit prediction of synoptic-scale fronts. Wea. Forecasting, 34, 1137–1160.

• Sanders, F., 1999: A proposed method of surface map analysis. Mon. Wea. Rev., 127, 945–955

• Schemm, S., Sprenger, M., & Wernli, H. (2018). When during Their Life Cycle Are Extratropical Cyclones Attended by Fronts?, Bulletin of the American Meteorological Society, 99(1), 149-165.

• Thomas, Carl M. and Schultz, David M., 2019, "Global Climatologies of Fronts, Airmass Boundaries, and Airstream Boundaries: Why the Definition of "Front" Matters" Monthly Weather Review Vol. 147, No. 2, pp 691, 1520-0493

• Thomas, Carl M. and Schultz, David M., 2019, "What are the Best Thermodynamic Quantity and Function to Define a Front in Gridded Model Output?" Bulletin of the American Meteorological Society Vol. 100, No. 5, pp 873, 1520-0

• Uccellini, L. W., S. F. Corfidi, N. W. Junker, P. J. Kocin, and D. A. Olson, 1992: Report on the surface analysis workshop at the National Meteorological Center 25–28 March 1991. Bull. Amer. Meteor. Soc., 73, 459–471.

   We included most reference into the text

**5 Co-Editor**

1. In order to make the paper of interest for a broader readership, the authors should include an application of their method that showcases the benefit of this method over existing ones (see also comment by reviewer 2).

   We added an Application where we investigate the connection between our detected fronts and extreme precipitation events as section 3.2.2. This application existed previously but our method allows us to evaluate at the much higher ERA5 resolution ($0.25°$ vs $2.5°$), while maintaining similar results as the previous study. This shows the benefit of our method with respect to the classical TFP methods, which are hard to apply on high resolutions.

2. As pointed out by reviewer 2, the subjectivity of fronts retrieved from manual surface charts can be problematic and I consider it essential the authors can alleviate this concern.

   We updated our Cross Section Plots, which now include THE and the second derivative which should be similar to the TFP to highlight that both our networks and the provided weather service fronts correspond well to the criterion used by traditional front detection algorithms ($\nabla(TFP) = 0$). Additionally we expanded the section where we describe how we handle the inherent bias of the labels, by allowing a deformation of said labels (See Section 2.2.4 and Fig. 5) We also updated our text accordingly. See chapter 3.2.1

3. Generally, the introduction should provide a broader overview over the existing literature on automated front detection and its applications. Also I recommend the authors expand the conclusion and put their method into a broader context.

   We expanded the introduction, and also the conclusion. We expanded the section on the Cross sections, which put our method in the context of both, fronts derived by weather services and with classical TFP methods.

4. Please make sure that all figures have appropriate labels (a), (b)... and that these are referenced in the figure caption.

   We did this

5. Clarify the labels in legend to Fig. 6. Also in Fig. 6b the legend should not be centered at 0, where the front is located. It should be moved to the edge as for the other panels.

   We corrected the labels and moved them to the edges.

6. The figures showing the extracted front features are difficult to interpret. At the very least they should include a geographical reference, such that, for example, in Fig. 2 a direct comparison can be made with the manual surface chart. Also plotting in the background some meteorological fields such as SLP and THE or similar would facilitate the interpretation of the detected fronts.

   We added a land/sea contour, as well as latitude and longitudinal axes to the plot. Additionally we show the THE in the same plots. In the provided video supplement we also show our detected fronts over THE on the global scale for January 2016 at a 1 hourly interval.

---

## Author Response (AR2)

**Minor Revision: Answers to Referees**

Stefan Niebler et al.

November 2021

**1   General**

1. We adjusted Figures 6,8,9, S2 and S3 to be more accessible regarding color blindness.

**2   Remarks from previous File validation**

1. Please provide a source of fig 2, 3 if you are the originator, please inform us.

   We added a reference in the corresponding figure captions, as the trademark in the images is rather small.

**3   Reviewer 1**

1. The authors have made a thorough revision of the manuscript based on the comments of the reviewers (many thanks!). The quality of the paper has improved significantly and I think it is now ready for publication.

   Thank you!

**4   Reviewer 2**

**4.1   General**

1. The authors present a revised version of their ML-based surface front detection. One certainly cannot dispute the usefulness of feature-based detection methods, but my reservation about whether manual surface analysis should guide the development of a next-generation automated feature-based method remains intact.

   We still think that manual surface analysis charts provide a sound ground truth for this method. Since we can show that the fronts derived from surface analysis charts on average have the thermodynamic properties, which would be found from a TFP-method, we use the surface analysis charts as ground truth.

2. The discrepancy between the traditional TFP-based methods, which arguably have their own shortcomings, and the presented method for emulating DWD and WCP fronts does not, in my opinion, indicate a particular weakness of the TFP-based methods, but must be considered in light of the weakness of the manual surface maps, which either fail to account for or erroneously indicate or displace relevant surface features otherwise correctly detected by the automated TFP-based method. The question remains as to what should be accepted as ground truth, and as stated earlier, I cannot recommend relying fully on manual analysis. This position is in contrast to several statements by the authors who continue to hold on to manual analysis as a ground truth and consequently continue to argue throughout the manuscript that traditional TFP-based methods are outperformed. The answer might simply be that the manual analysis is erroneous in many cases, and the ML-method has learned the bias while the TFP-based method – in fact – outperforms both. It is simply a fundamentally different viewpoint.

   Actually, we show in a comparison that the fronts determined by the weather services (and also by our network method) exhibit the characteristics in terms of temperature gradients etc., which one would expect from a front identified using the TFP-method. Thus, it is not a different viewpoint, since fronts determined from both (and different) methods should usually agree very well.

3. Even though the authors have improved their comparison between a TFP method and their new method, I still think the comparison is incorrect. A proper choice for a baseline method must always be seen relative to the ground truth. Here, manual surface charts are used as ground truth, which are drawn based on several variables at several heights, and as baseline a method is chosen which uses one variable at one height and was never developed with the specific goal to reproduce surface charts. As already mentioned, I recommended removing this comparison, since it is not necessary for the publication. Only the comparison with an earlier ML method would make sense. However, since even the authors of this study argue in their reply that they are not able to handle the code provided by of one of the earlier ML methods, I am concerned about the reproducibility of these studies. At the end of this document, I recommend another ML method that uses the same ground truth – maybe this code is more user friendly and can serve as the baseline method the author wish to have.

   We do not think that a comparison with the mentioned ML method is really meaningful. We did not find the provided code base, and the results do not seem to be very robust.

4. Nevertheless, in their revised introduction, the authors have addressed aspects of this discussion, but the need for labeled training data is so central to their method, there is basically no other option for the training

of the ML method but to accept the author decision and evaluate the manuscript considering their viewpoint.

As mentioned above, we still think that manual surface charts are a sound ground truth, therefore we use them in our study.

5. To me then, the automated method gives us gridded front data that might be useful for meteorological research related to phenomena associated with the passage of surface fronts. The presented example, a confirmation of an earlier studies that addressed the question of front-related extreme precipitation events, leaves unfortunately the question open of what exactly can be learned using ML-based methods given that the authors basically show that the method reproduces exactly what was previously found using a TFP-based method. Recommendation: It would be helpful to at least give some indication at the end of this section of what exact new physical insight can now be generated with the new method that could not be generated before.

It is stated in the text that this evaluation was also carried out using much smaller radii on the high resolution ERA5 data. Some new results are presented in the supplement. Such an evaluation can hardly be carried out using a TFP method, since these methods have problems with high resolution data as we could see in our investigations using the baseline method.

6. The climatological application is otherwise a very nice example that would motivate a section on the issue of explainability of data-driven methods. While the presented method produces climatological patterns in agreement with previous findings, it is beyond that capable of splitting different front types in a clear manner. Of particular interest would be to understand what variables are key for the learning process and if the climatological patterns would look different if only trained on a single variable. A particular strength of the ML method could be to use a low number of input features to reproduce manual analysis. Again, to me, however the results seem to result from the combination of various input channels, while traditional methods often rely on a single variable which seem to be not sufficient to separate different front types. Layerwise backward propagation might be a simple way of showing what variables allow the network to develop this ability. Recommendation: It would be useful to give some indications in this direction at the end of the corresponding section.

We have carried out such an investigation, but in our case this was not successful. Actually, it is well known that such an attribution method does not always work.

7. In the summary it is argued that the method can also be applied to higher-resolution data. I think this is not the case. To make the method mesh independent, the input training data would need to be converted to continues space and training would need to be performed in continues space as

is done in random feature methods or eventually also in FFT-space. There is something to be said here about mapping between Banach spaces.

It is stated in the manuscript that there are no principal obstacles to apply this method for high resolution data. We did not state that the method is mesh independent. It might be that the network (pre-) trained on ERA5 data must be further trained for the use on high resolution data.

**4.2   Introduction**

1. The authors are encouraged to add more reference to their statements relating fronts to, for example, wind gusts or extreme weather.

We added references Catto and Dowdy (2021), Catto et al. (2015) and Martius et al. (2016) to the manuscript at Line 20

**4.3   Some ML related questions:**

1. Is the Batch normalization really needed? Usually, it accelerates the training process and additionally improves the skill. However, from a theoretical viewpoint, it is unclear why this is case and thus it might not be needed in this particular application.

While it is certainly possible to construct and train networks without batch normalization (BN), it has a number of favorable properties that improve results significantly: Aside from faster training, the main benefit of batch normalization is an increase in generalization performance, i.e., deep networks trained with BN are less prone to overfitting. This effect has been verified empirically many times, see for example:

Johan Bjorck, Carla Gomes, Bart Selman, Kilian Q. Weinberger Understanding Batch Normalization NeurIPS 2018

https://proceedings.neurips.cc/paper/2018/file/36072923bfc3cf47745d704feb489480-Paper.pdf

From a more theoretical point of view, there are connections to increasing the margin of the classifier. See for example:

Jure Sokolic, Raja Giryes, Guillermo Sapiro, Miguel R. D. Rodrigues Robust Large Margin Deep Neural Networks https://arxiv.org/pdf/1605.08254.pdf IEEE Transactions on Signal Processing, 2017

- Why BN in U-Nets?

The architecture used in our paper is a U-Net, which is a type of network that uses "skip-connections" in order to reduce training problems. One might think that the introduction of skip connections means that batch normalization is not very useful any longer, as these architectures are less prone to vanishing gradients and related problems. However, there is empirical and theoretical evidence against this, see for example:

A. Labatie: Characterizing Well-Behaved vs. Pathological Deep Neural Networks. ICLM 2018. (nicely summarized here: https://towardsdatascience.com/its-necessary-to-combine-batch-norm-and-skip-connections-e92210ca04da )

The author shows that deeper architectures are prone to confinement of the data signal to low-dimensional subspaces (the singular values of the Jacobian drop quickly; one could call this vanishing dimensionality), and show that only a combination of BN and skip-connections can reliably counteract the problem.

In summary, BN is usually included in most modern architectures as folklore-based "best-practice"; literature gives us, nonetheless, rather strong evidence that this is highly useful from both an empirical and theoretical perspective. Aside from improving numerical conditions (gradients, singular value spectrum of the Jacobian) it also improves generalization performance / reduces overfitting tendencies.

2. Why is the drop-out chance set to 0.2? Is there any over-fitting without it? How does this relate to the problem of choosing arbitrary thresholds? I recommend a brief discussion of the sensitivity.

Lagerquist et al. found that a high dropout worked well to avoid overfitting in their case (0.25 and 0.5). It does not appear that our network is overfitting at this point, which is why we did not use a higher dropout. Potentially a lower dropout may work as well.

3. Why did you choose 3 drop-out layers and avg. pooling steps in your U-Net architecture and not less or more?

Using less layers performed worse in our first tests. Using 4 encoding and decoding blocks would exceed the memory of the used GPUs. This study is however not about finding the optimal network architecture, but to apply the network to a meteorological problem.

4. Why are the number of channels changing from 330 to 64 after the first encoding block, but for all further encoding it increases by a factor of two?

With this first encoding block we tried to learn some kind of embedding of the variables. It slightly improved results but it might not be necessary for the whole cause.

5. Reference for U-Net should also be given to Shelhamer et al. 2016 (doi: 10.1109/TPAMI.2016.2572683) We added this. See Line 98

6. L. 345, how did you determine the deformation factor of k=3? Shouldn't the choice be tested against randomness in some way? How, as before, does this choice relate to the problem of choosing arbitrary thresholds? A common weakness of traditional methods.

Basically we were looking at the width of the results from k=0. if we choose $k = 1$, we obtain double lines, as it cannot fully cover the deviation. k=3 works quite well as it is appears to reliably cover the width of the bias.

We did not test $k > 3$, as $k = 3$ already works. Yes this value is chosen arbitrarily, but this is not critical for the detection of fronts rather than the placement/width of the fronts related to the label bias.

**4.4 Section 2.2.4**

1. The authors are encouraged to add more reference to their statements relating fronts to, for example, wind gusts or extreme weather.

   We do not understand where in this section this should be included, as it is not the topic of the section. As stated before we added such references into the introduction.

2. Several previous studies have questioned the usefulness of front lines in general and for the use in next-generation front detection methods. These studies rather recommend using frontal regions or frontal volumes.

   Evaluating the temporal evolution of a front at a surface station, it can be often seen that the surface front consists of a very narrow line, as e.g. can be seen in the surface pressure. We added an image from the passage of storm Niklas at the weather station in Mainz in 2015, which shows clearly the small extent of the surface front (pers. comm. P. Reutter, https://www.ipa.uni-mainz.de/wetter-alt/wetterbesonderheiten/)

[Figure]

3. Is all of what is done in this section needed simply to obtain front lines?

   Yes this section was mainly done to create lines. Furthermore this is a problem of available data as there is no labeled datasets of wide lines. And in the case of simply expanding the given frontal lines, we propose that one can simply widen the results of the network as well.

**4.5 Section 2.3**

1. I recommend removing this section and the corresponding comparison in Section 3.1.1. Also, it is noted that only midlatitude fronts are included

for the TFP method, but in Section 3.2.2. the opposite is done.

We do not agree that this section should be removed. We did not quantify the TFP method outside the midlatitudes, which we agree would not be a fair comparison. However we do not agree that not showcasing the shortcomings of a method and comparing how our method fairs in those regions should not be done.

In 3.2.2 the TFP method was applied by Catto and Pfahl. We only used our network in this section.

In 3.1.2 (which might be what you mean) the fronts outside the midlatitudes are not used in the quantification. They are used in the discussion part of the section, but we still believe that it is fair to also highlight the cases where a certain method does not work correctly and show if a proposed method can perform better there.

**4.6  Section 2.4.3**

1. Even though POD and SR are intuitive measures, I recommend to better explain the meaning of nmws and nws. The latter is "the count of all provided fronts" the former "all fronts that could be matched". To what does provided refer to (provided by whom)? What is a front that is provided but cannot be matched?

   Line: 412 "We define nMWS as the count of fronts provided by a weather service,"

   What is a front that is provided but cannot be matched?: Such a front is irrelevant to the evaluation. However, such a front is a front that is provided by the weather service, that did not fulfill the matching criterion (The criterion mentioned in the section starting at line: 396)

2. Fig. 6 is missing a color bar for the gray shading.

   We added the color bar to the figure

3. Fig. 6 The yellow class is labelled as "no class" but there seems to be no yellow label in the figure.

   There is at the center image bottom row. Albeit it is only a very small part.

**4.7  Section 3.2**

1. Overall, I am afraid I do not understand the purpose of this section. Is it about showing that DWD and WCP fronts have gradients?

   Yes. It further shows that both the results of the network as well as the DWD and WCP fronts exhibit the same characteristics as those that would have been found by a TFP method. This shows that the surface fronts identified by the weather services are a reasonable ground truth for learning to detect fronts.

2. Fig.9: What is the variance for the shown averaged values for each line and are the differences between the methods within or outside, for example, the range given by -/+ two times the standard deviation of the sample that went into the averaging for each method?

We absent from a statistical test as we are averaging on rather large temperature differences. The standard deviation itself for the equivalent potential temperature for example lies at approximately $10K$ to $15K$. This section is not intended to statistically determine the difference between weather service provided and network detected fronts, but rather to show that both generate fronts that are in line with the expected behaviour (e.g. TFP criterion, temperature gradients, ...)

3. The lines all look very similar to me and may not significantly be different from each other.

This section is intended to show that the results of our trained network correspond to the expected behaviour of the different types of fronts. It is to be expected that these results correlate with the cross sections of the weather service fronts, as long as both results truly show fronts.

4. In all honesty, this does section does not add much to the paper. This section should be removed as the paper can be published without this information.

We do not agree. We believe that this is a very important section, as it shows that the surface fronts of the weather services are a good label, as they appear to fulfill the criterion used by the TFP methods rather well. Additionally the detected fronts also agree very well with this criterion.

**4.8 Section 3.2.2**

1. I am afraid I do not support the usage of an attribution measured that uses an attribution radius defined in terms of degrees. I would assume that 2.5 degrees correspond to a different area/distance at different latitudes so you will attribute less precipitation to fronts at higher latitudes, don't you?

Yes, this is true. However, we tried to stay as close to the other paper as possible. In fact a $km$ based attribution may be more accurate however as you stated it would most like result in a higher matching rate, as the attribution radius in the higher latitudes would increase. This is also an advantage of our method, as it is applicable to ERA5 we can in future work use a $km$ based attribution ratio to research connection of fronts and other events more accurately.

2. Not sure if the difference between fr and a(fr) is fully clear. Is the first the number of fronts at a grid point and the second a probability? What do you mean by "grid point p is associated with a front" other than "a front occurs at p"?

We added some text to make this more clear: A "grid point p is associated with a front "a(fr)"", if it resides within a 2.5° distance of any front. On the other hand " a front occurs at p" means that a front is located at this exact grid point. The same distinction for extreme precipitation. We extended the text in the listing at Lines 676 and 679

3. Fig. 10: Maybe I missed it but why are the polar regions not shown?

Because the other paper did not do this as well. Plus the resolution of ERA5 becomes very inaccurate the more pole ward we evaluate.

4. Fig. 10-12: Some words in the title of the figures are capitalized others not.

we made them all lower case

**4.9 Literature**

1. The authors may consider the following paper, which appears to target the same ground truth but uses a random forest method. I guess that this is the baseline method the authors are looking for. Bochenek, B.; Ustrnul, Z.; Wypych, A.; Kubacka, D. Machine Learning-Based Front Detection in Central Europe. Atmosphere 2021, 12, 1312. https://doi.org/10.3390/atmos12101312

Thank you for this link. We will add it to our literature. However, we do not think that at this stage we should compare against this method, as we did not find provided code base nor do the results appear very robust, regarding Table 3. So we do not believe that it provides a good baseline to compare against. We might do so in a future work. We added it with some text at Line 85.

---

## Author Response (AR3)

**Minor Revision: Answers to Co-Editor**

Stefan Niebler et al.

December 2021

**1 Co-Editor**

1. The maximum permitted level of sectioning in WCD is three as you can see from the "manuscript composition" guidelines (https://www.weather-climate-dynamics.net/submission.html#manuscriptcomposition). Please correct this in sections 2.4 and 3.2.

   We corrected this in the mentioned sections. Some text had to be adjusted to accommodate for these changes (e.g. texts regarding the content of certain subsections etc.)

2. The analyses presented in section 3.2.2 (Correlation to extreme precipitation events) need detailed explanations of the method. These should be placed in the methods section instead of being presented along with the results. Given the length of the paper in general and the methods section in particular I recommend to move this to the supplement. This will make for a more compact manuscript, though it will require some minor rewriting of Section 3.2.2 to ensure that it remains comprehensible.

   We moved most methodological parts into the supplement (e.g. statistical test, mathematical definitions, ...). We adjusted the chapter accordingly. We also reduced the sectioning depth by setting the whole section as chapter 3.3 instead of 3.2.2. As a result we also adjusted the introductory text for chapters 3.2 and 3.

3. The conclusion contains one excessively long paragraph. Please consider restructuring into several paragraphs for better readability. The same applies to the long paragraphs starting on lines 524 and 601 of the non-track-changed manuscript.

   We split these paragraphs into several smaller paragraphs, where we considered it appropriate.

4. Please screen the manuscript for typos and grammatical errors.

   We reread and updated the manuscript to reduce typos and grammatical errors. We also added references to software packages used during our evaluation.

5. L37: It is not clear what you mean by "reference data-sets". Suggest to remove the word reference.

   We removed the word "reference"

6. L106: two should not be in bold face

   "two" is no longer bold face